# Unscrambling fluorophore blinking for comprehensive cluster detection via photoactivated localization microscopy

René Platzer[1,3], Benedikt K. Rossboth[2,3], Magdalena C. Schneider [2], Eva Sevcsik[2], Florian Baumgart[2], Hannes Stockinger [1], Gerhard J. Schütz [2], Johannes B. Huppa [1✉] & Mario Brameshuber [2✉]

Determining nanoscale protein distribution via Photoactivated Localization Microscopy (PALM) mandates precise knowledge of the applied fluorophore's blinking properties to counteract overcounting artifacts that distort the resulting biomolecular distributions. Here, we present a readily applicable methodology to determine, optimize and quantitatively account for the blinking behavior of any PALM-compatible fluorophore. Using a custom-designed platform, we reveal complex blinking of two photoswitchable fluorescence proteins (PS-CFP2 and mEOS3.2) and two photoactivatable organic fluorophores (PA Janelia Fluor 549 and Abberior CAGE 635) with blinking cycles on time scales of several seconds. Incorporating such detailed information in our simulation-based analysis package allows for robust evaluation of molecular clustering based on individually recorded single molecule localization maps.

[1] Institute for Hygiene and Applied Immunology, Center for Pathophysiology, Infectiology and Immunology, Medical University of Vienna, Vienna, Austria. [2] Institute of Applied Physics, TU Wien, Vienna, Austria. [3] These authors contributed equally: René Platzer, Benedikt K. Rossboth. ✉email: johannes. huppa@meduniwien.ac.at; brameshuber@iap.tuwien.ac.at

PALM is a single-molecule localization microscopy (SMLM) technique devised to resolve structures below the diffraction limit. It relies on stochastic photoswitching of fluorophores, either irreversible or reversible, between a blue-shifted fluorescent or dark state and a red-shifted or bright fluorescent state[1,2]. Among photoswitchable fluorescent proteins (FPs) utilized so far in PALM are Kaede[3], PAmCherry[4], Eos-based proteins[5], Dronpa[6], mClavGR-based proteins[7,8], and variants of GFP such as photoactivatable GFP[9] (PA-GFP) and photoswitchable-CFP2[10,11] (PS-CFP2) (reviewed in ref. [12]). Reversible photoswitchable organic dyes (reviewed in refs. [13,14]) have been mostly employed for (direct) stochastic optical reconstruction microscopy ((d)STORM)[15,16] applications, but are in many cases not compatible with live-cell labeling and PALM. Only a few photoactivable organic fluorophores are described, which are based on uncaging of rhodamine dyes[17–19] or fluorescein[20,21]. Photoswitching only a minor fraction of fluorophores into the active state gives rise to well-separated single-molecule signals, which are localizable with a precision primarily determined by the signal-to-noise ratio[22].

Numerous studies have addressed protein clustering within the plasma membrane using various photoswitchable or photoactivatable FPs (reviewed in ref. [23]). However, the observation of repetitive on-off blinking challenged data interpretation, likely giving rise to false-positive cluster detection due to fluorophore overcounting[24]. Notably, mEOS2 exhibited non-negligible light-induced fluorescence recovery from dark states[25] which was implicated in the recording of erroneous protein assemblies[24]. In contrast, blinking of PS-CFP2 was considered less pronounced[24,26] and a high fraction of photoswitched PS-CFP2 molecules was reportedly bleached irreversibly upon high-powered 488 nm irradiation[10,27]. Still, to this date, the observation of reversible PS-CFP2 blinking cycles before terminal photobleaching[24,26,28,29] has remained a challenge not only for quantitative PALM approaches[29–32] but also for more general conclusions relating to functionally relevant protein clustering and protein co-localization.

In this study, we devise and test a widely applicable experimental and analytical platform to determine the blinking signature of photoswitchable and photoactivatable fluorophores for comprehensive protein cluster assessment in single cells. Employing common imaging conditions[27] we characterize single PS-CFP2, mEOS3.2, PA Janelia Fluor, and Abberior CAGE635 molecules, chosen here as representative fluorophores. Individual molecules of all probes tested are detected on average more than four times with blinking cycle times in the order of seconds. Monte Carlo simulated detections of randomly distributed fluorophores featuring blinking properties of the respective molecule yield considerable apparent clustering, which could, as we demonstrate here, be quantitatively accounted for when evaluating PALM-generated localization maps. Our study highlights the need to reliably detect and experimentally optimize the blinking properties of photoswitchable and photoactivatable fluorophores prior to their use in PALM when aiming for a quantitative characterization. Our overall approach serves to precisely determine fluorophore-specific blinking parameters for reliable cluster evaluation of individually recorded localization maps.

## Results

### Platform development to quantitate fluorophore blinking. We selected for this study PS-CFP2 and mEOS3.2 as photoswitchable model fluorophores in view of their wide use in PALM and their reportedly low to moderate blinking tendencies[24,27], and

Abberior CAGE635 (CAGE635) and the recently developed PA Janelia Fluor 549 (PA-JF549) as photoactivatable organic dyes. To be able to determine fluorophore-specific blinking properties with single-molecule resolution, we site-specifically coupled PS-CFP2, mEOS3.2, PA-JF549, and CAGE635 via a biotin modification to monovalent and recombinant streptavidin (mSAv*-3xHis$_6$, Supplementary Fig. 1), which was then anchored by means of its three histidine tags to a gel-phase planar glass-supported lipid bilayer (SLB) consisting of 1,2-dioleoyl-sn-glycero-3-[(N-(5-amino-1-carboxypentyl)iminodiacetic acid)succinyl] (nickel salt) (DGS-NTA(Ni)) and 1,2-dipalmitoyl-sn-glycero-3-phosphocholine (DPPC) (Fig. 1a). Specifics concerning the generation of mSAv*-3xHis$_6$, its complex with PS-CFP2, mEOS3.2, PA-JF549, or CAGE635, and the protein-functionalized SLBs are described in the "Methods" section and Supplementary Figs. 1 and 2. This platform supported 2-dimensional imaging in total internal reflection (TIR) mode, as carried out in many PALM studies.

The high melting temperature (41 °C) of the matrix lipid DPPC[33] afforded continuous observation of SLB-immobilized fluorophores. Furthermore, the use of SLBs prevented non-specific fluorophore binding to the glass surface. False-positive detections caused by trace amounts of (partly) hydrophobic buffer- or lipid-derived dyes[34] intercalating with synthetic lipid bilayers (Supplementary Fig. 2B) were minimized by two-color co-localization between the single-molecule signals of the fluorophore under investigation and the dye-conjugated mSAv*-3xHis$_6$ platform. To this end, either the green-emitting dye Alexa Fluor 488 (AF488)—for the characterization of CAGE635—or the red-shifted dye Abberior STAR635 P (STAR635)—for the characterization of PS-CFP2, mEOS3.2 and PA-JF549—were site-specifically and quantitatively conjugated to an unpaired cysteine residue engineered within the biotin-binding subunit of mSAv*-3xHis$_6$ (mSAv*-STAR635 or mSAv*-AF488, Fig. 1a, Supplementary Fig. 1A–D). A two-color data acquisition protocol with sequential excitation of both fluorophores was applied to select specifically for the photoswitchable or photoactivatable fluorescent molecule under investigation co-localizing with mSAv*-STAR635 or mSAv*-AF488. Figure 1b shows a representative overlay of an image recorded in the red (mSAv*-STAR635) color channel and single-molecule detections determined for individual PS-CFP2 in the green color channel. Single-molecule signals of all fluorophores employed were well discriminable from background noise and were detected and localized by the image analysis tools employed with a signal-to-noise ratio ranging—depending on the fluorophore under investigation—between 30 and 50 (Supplementary Fig. 2A). Importantly, applying the same imaging conditions without the presence of any fluorophore resulted in only a few detections of non-fluorophore related background signals (Supplementary Fig. 2B).

In control experiments conducted with SLBs solely decorated with mSAv*-STAR635, only 2.6 ± 0.33% (mean ± SEM) of detections in the green channel colocalized with mSAv*-STAR635 visualized in the red channel, indicating a 38-fold reduction in detecting false-positive signals by applying the two-color co-localization protocol. Single-molecule tracking of individual mSAv*-STAR635 molecules testified to its quasi immobile state with a diffusion coefficient $D < 10^{-4}$ μm$^2$/s (Supplementary Fig. 2C), rendering the platform suitable for the observation of the very same fluorophore over timescales of seconds. Figure 1c shows two representative intensity traces of PS-CFP2 signals recorded in a PALM experiment. Of note, emissive behavior varied substantially between different molecules and ranged from one-frame detections to repeated detections over multiple frames interrupted by several non-emissive gaps.

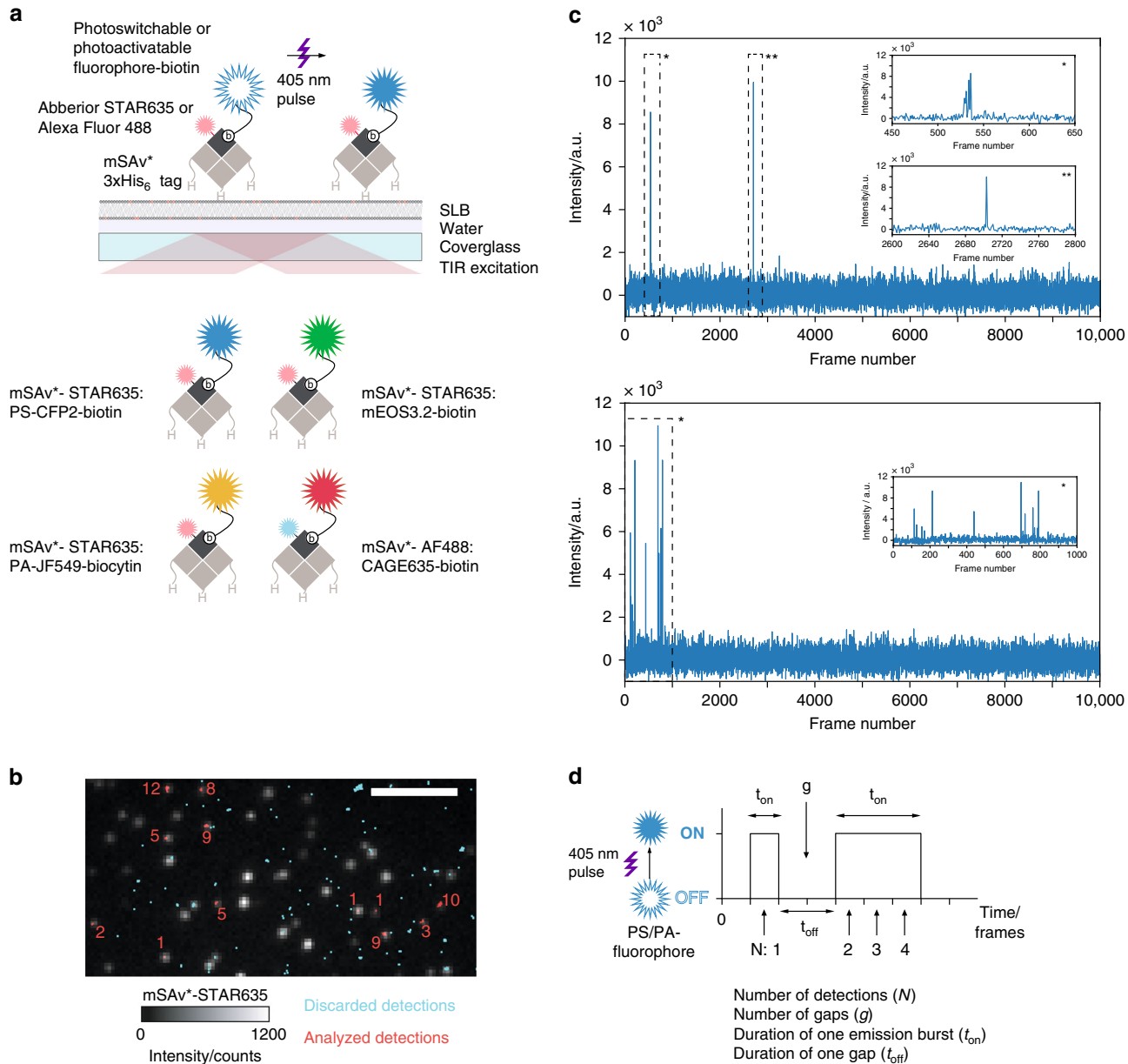

**Fig. 1 Imaging platform to quantitate blinking parameters of recombinant photoswitchable (PS) or photoactivatable (PA) fluorophores. a** PS/PA fluorophore-mSAv*-STAR635 (or-AF488) conjugates were anchored to immobile SLBs containing 99.5% DPPC and 0.5% DGS-NTA(Ni). PS/PA fluorophores were identified by means of co-localization with mSAv* site-specifically conjugated to STAR635 or-AF488, which allowed for precise acquisition of blink statistics in TIR mode on a single-molecule level (top). All four characterized PS/PA fluorophores, i.e., PS-CFP2, mEOS3.2, PA-JF549, and CAGE635, are displayed together with their respective label used for co-localization analysis (bottom). **b** Two-color co-localization of STAR635 and PS-CFP2 enabled discrimination of false-positive detections as shown in the images acquired for mSAv*-STAR635 (monochrome color channel) and from consecutive PALM of single PS-CFP2 signals (red and cyan colored detections). The total number of PS-CFP2 detections per analyzed signal are indicated in red. Scale bar is 4 µm. **c** Representative intensity traces acquired in 10,000 image frames for a single PS-CFP2 molecule exhibiting two bursts of detections (upper panel) and another exhibiting multiple burst (lower panel). Inserts provide a higher temporal resolution of indicated bursts. **d** Four parameters were extracted from single-molecule traces: (i) the total number of detections ($N$), (ii) the number of off-gaps ($g$) occurring within single traces, (iii) the duration of each emission burst ($t_{on}$), and (iv) the duration of each off-gap ($t_{off}$).

## Determining the blinking parameters of PS-CFP2, mEOS3.2, CAGE635, and PA-JF549.

We analyzed the temporal emission patterns of single PS-CFP2, mEOS3.2, PA-JF549, and CAGE635 molecules with regard to four parameters affecting the interpretation of localization maps (Fig. 1d): (i) the total number of detections per molecule within the whole imaging sequence ($N$), (ii) the number of gaps within a single PS or PA fluorescent molecule trace ($g$), (iii) the duration of each emissive state ($t_{on}$) and (iv) the duration of each gap ($t_{off}$). Of note, high values in

$N$, $g$, and $t_{on}$ can in principle be adequately accounted for by simple merging strategies as long as values of $t_{off}$ are low.

Where indicated, we included a chemical fixation step involving 4% paraformaldehyde (PFA) prior to fluorophore recording as this reflects experimental conditions chosen in the majority of SMLM-based image acquisitions (Fig. 2). For example, when applying a 488-nm excitation power density of 3.0 kW cm$^{-2}$ for 2 ms exposure at a frame rate of 167 Hz, a single PS-CFP2 molecule was detected on average 4.08 times over the

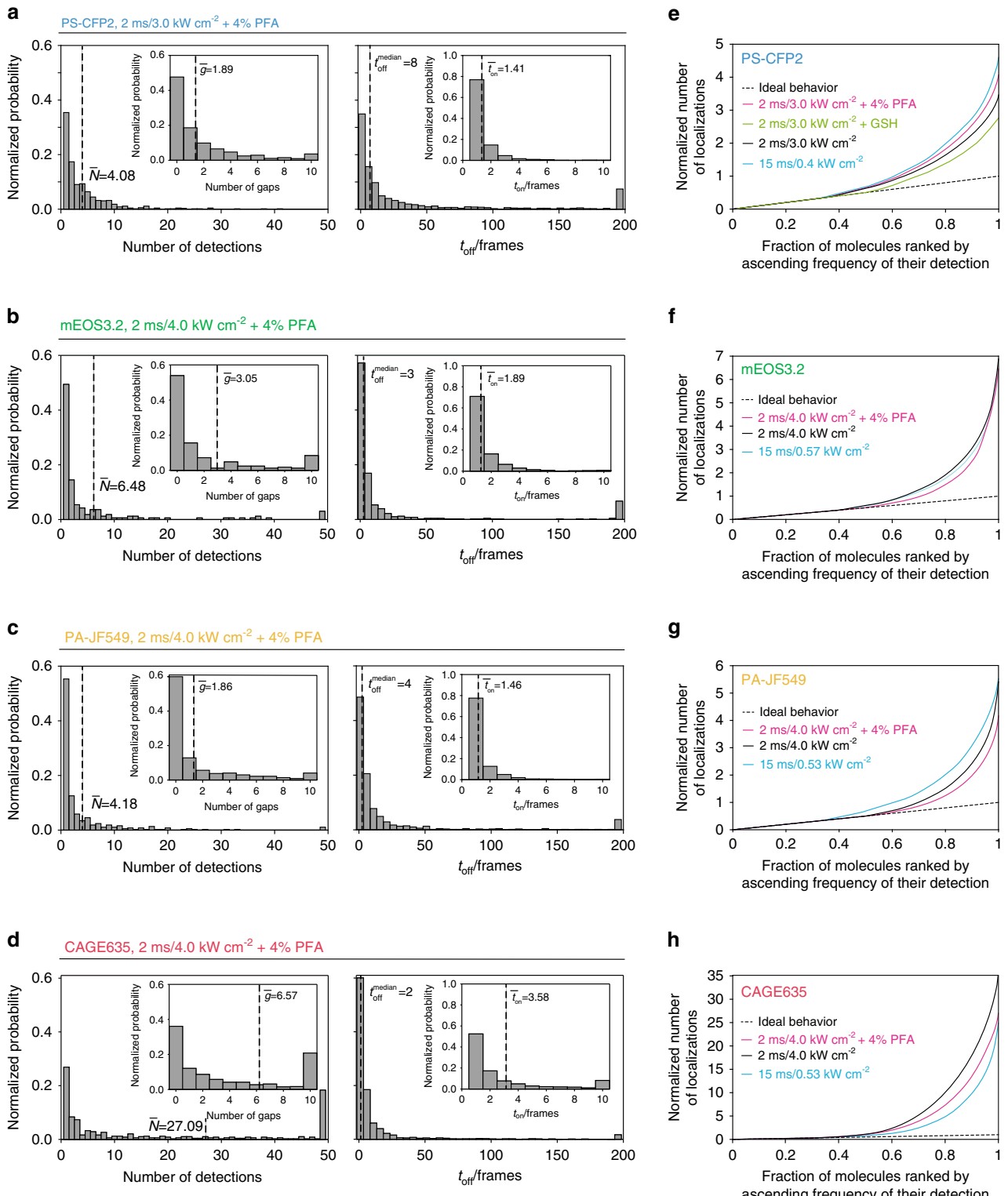

**Fig. 2 Quantitation of PS-CFP2, mEOS3.2, PA-JF549, and CAGE635 blinking characteristics treated with 4% PFA and at high power density. a–d** Normalized histograms illustrate the four parameters described in (Fig. 1d) determined for two photoswitchable fluorescent proteins (PS-CFP2 and mEOS3.2) and two photoactivatable organic dyes (PA-JF549 and CAGE635) treated with 4% PFA and at indicated power densities with an illumination time of 2 ms. As indicated, numbers within the histograms indicate the mean or median value of the respective parameter. In total, $n = 720$ (PS-CFP2), 166 (mEOS3.2), 375 (PA-JF549), and 491 (CAGE635) molecules have been recorded on 77 (PS-CFP2), 56 (mEOS3.2), 72 (PA-JF549), and 156 (CAGE635) independent lipid bilayer positions. **e–h** Quantification of PS-CFP2 (**e**), mEOS3.2 (**f**), PA-JF549 (**g**), and CAGE635 (**h**) blinking by plotting the normalized cumulative sum of detections against the fraction of molecules recorded. The black dotted diagonal indicates a hypothetical ideal scenario, in which each PS or PA fluorescent molecule gives rise to one single detection. Deviations from the diagonal indicate multiple detections per PS or PA fluorescent molecule. All curves are normalized to the mean number of detections for a given setting. A statistical comparison of the four parameters at indicated imaging conditions and between investigated fluorophores is provided in Supplementary Tables 1 and 2.

course of 10,000 frames (Fig. 2a, left). Sixty-five percent of molecules were detected more than once in the whole image sequence; about 25% of all molecules appeared more than five times with a maximum of 41 detections. A similarly broad distribution was observed for the relative number of gaps, $g$, occurring within individual PS-CFP2 traces: 52% of PS-CFP2 molecules showed at least one gap; we found a maximum number of 37 gaps (Fig. 2a, insert left panel). The duration of emissive states was on average 1.41 frames (Fig. 2a, insert right panel). About 20% of detections resulted from signals localized in consecutive frames, and only a minor fraction of 1% was detected in five or more consecutive frames. For molecules undergoing emission gaps, we further analyzed the distribution of $t_{off}$ (Fig. 2a, right). The median off-time was eight frames. While most molecules showed off-times shorter than 50 frames, the distribution featured an extended tail towards long $t_{off}$ values: 8% of the observed off-times ranged between 200 and ~9000 frames (full distribution not displayed for improved graph clarity, all events merged into the last bar), which would be too high to be accounted for by simple event merging without losing information with regard to real clustering.

For mEOS3.2, we applied 532-nm excitation and similar settings as employed for PS-CFP2 recordings. On overage, a single mEOS3.2 molecule was detected 6.48 times with 3.05 gaps per trace, 1.89 frames duration of the emissive state ($t_{on}$), and a median off-time of 3 frames (Fig. 2b). Experiments involving the organic dye PA-JF549 were performed with the settings employed for mEOS3.2, and yielded distributions similar to that of PS-CFP2 (Fig. 2c), however, with a twofold reduced off-time. The red-shifted organic PA fluorophore CAGE635 was excited at a wavelength of 647 nm; otherwise, the same settings used for the characterization of PA-JF549 were applied. Blinking of CAGE635 was considerably more pronounced when compared to that of all other fluorophores tested, yielding on average 27.09 detections, 6.57 gaps, and 3.58 frames duration of the emissive state ($t_{on}$) per molecule, and a median off-time of two frames (Fig. 2d).

**Blinking properties of PS and PA fluorescent molecules are affected by experimental conditions.** PALM experiments described above were carried out following paraformaldehyde sample treatment, i.e., under conditions employed for imaging chemically fixed cells. Since sample pretreatment may not always be preferred, we characterized fluorophore blinking also in the absence of sample fixation (Supplementary Fig. 3). Of note, for all fluorophores investigated distributions of blinking parameters were not strongly affected by omitting PFA fixation (Supplementary Table 1).

Super-resolution imaging protocols differ in applied acquisition rates, which typically range from 33 Hz[35] to 250 Hz[27], and also with regard to the excitation intensity of the imaging laser. Since the laser light employed for excitation drives at least in part transitions between bright and dark states, we sought to identify experimental settings that reduce overcounting.

To this end, we examined how blinking changes when lowering the excitation power 7.5-fold while keeping the energy density at the sample constant by increasing the illumination time accordingly. As shown in Supplementary Fig. 4, the mean number of detections per PS-CFP2 and PA-JF549 increased by 13% and 30%, respectively, mainly due to an increased $t_{on}$. With regard to the remaining parameters, we did not observe significant differences for all fluorophores investigated (Supplementary Table 1).

For acquisitions of PS-CFP2, the least bleach-resistant fluorophore studied here, we also increased the excitation power density from 3.0 to 15 kW cm$^{-2}$ in an attempt to bleach every PS-

CFP2 molecule within the first image frame after photoconversion. As shown in Supplementary Fig. 5, the number of background detections with comparable signal-to-noise ratios increased, and we were hence unable to clearly discriminate between detections of single PS-CFP2 molecules and of background signals. The emergence of background signals was consistent with their time of appearance: while PS-CFP2 molecules were predominantly detected upon application of the photoactivation laser pulse at the beginning of a PALM recording, background signals were recorded with same frequency throughout the image stream (Supplementary Fig. 5B). Consequently, reliable blinking statistics could not be raised at this excitation power density due to the emergence of unrelated diffraction-limited fluorescence signals which cannot be discriminated from that of single PS-CFP2 molecules. This assessment was supported by the broad distribution of measured $t_{off}$ which reflects the appearance of background signals at random time points throughout the image sequence (Supplementary Fig. 5C).

In some applications, the fluorescent protein tag might be exposed to the cytoplasm, which features a reducing redox potential attributable to reduced glutathione[36] (GSH) present in millimolar concentrations. Since fluorophore bleaching and blinking is linked to oxidation of free radicals, we analyzed the emission characteristics of PS-CFP2 in the presence of 5 mM glutathione. We chose PS-CFP2 as a potential candidate due to its previously described use for labeling the intracellular part of the T-cell antigen receptor-associated CD3ζ chain (TCR-CD3ζ)[27,37]. The mean number of detections per PS-CFP2 molecule dropped to 2.77 with a concomitant decrease in the number of gaps per PS-CFP2 molecule (Supplementary Fig. 5D). While the duration of the emissive states remained unchanged compared to non-reducing conditions, the median duration of gaps increased from 10 to 18 frames.

A direct means of revealing single-molecule blinking behavior is plotting the normalized cumulative sum of detections versus the fraction of molecules recorded (see the section "Single-molecule blinking analysis" under Methods). In an ideal scenario, that is, if a fluorophore were only detected once, this relation yields a straight line with a slope of 1 (Fig. 2e, black dotted line). Each additional detection per molecule results in a deviation from this diagonal towards a higher number of detections per fluorophore. The plots obtained with PS-CFP2 (Fig. 2e) differ substantially from the diagonal; their maxima of about 2.8 (2 ms exposure at 3.0 kW cm$^{-2}$/167 Hz frame rate, 5 mM GSH; green line), 3.5 (2 ms exposure at 3.0 kW cm$^{-2}$/167 Hz frame rate, black line), 4.6 (15 ms exposure at 0.4 kW cm$^{-2}$/53 Hz frame rate; cyan line), and 4.1 (2 ms exposure at 3.0 kW cm$^{-2}$/167 Hz frame rate and after PFA fixation; magenta line) are equivalent to the mean number of detections per PS-CFP2 molecule. Depending on the imaging conditions, 50% (green line), 37% (black line), 35% (magenta line), or 26% (cyan line) of the PS-CFP2 molecules were detected only once, which, taken as a whole, accounted however for only 18%, 11%, 16%, or 5% of all detected signals. Hence, a total of 82–95% of PS-CFP2 detections arose from molecules which had appeared multiple times. Even more pronounced blinking properties were observed for all other fluorophores investigated in this study as can be inferred from the normalized cumulative sum of detections (Fig. 2f–h).

**Blinking distorts SMLM cluster maps.** Reappearance of one and the same blinking molecule gives rise to false-positive protein clusters, especially when fluorophores are no longer mobile after chemical fixation. To illustrate how blinking affects data interpretation, we juxtaposed a randomized distribution of immobile fluorophores with its corresponding localization map (Fig. 3a),

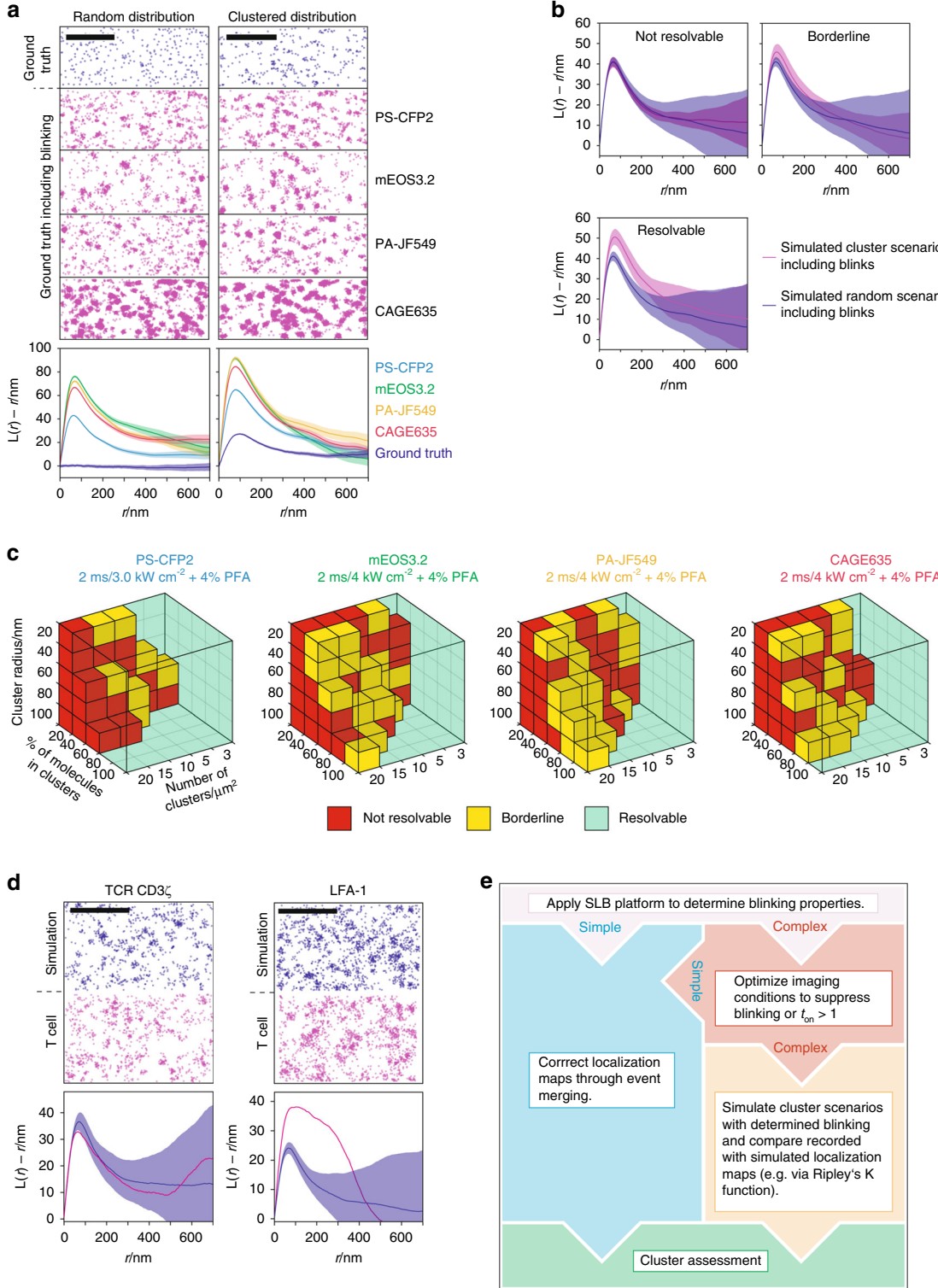

which was simulated based on the blinking characteristics of the respective photoswitchable or photoactivatable fluorescent molecule (determined under 4% PFA conditions as summarized in Fig. 2). The emergence of artificial clusters became evident already upon unaided inspection: a single blinking molecule gave rise to a cloud of detections with a lateral extension in the order of their positional accuracy. Without fluorophore blinking, Ripley's K analysis, which is one of several commonly applied analytical methods for cluster description in SMLM[38], yielded a constant value (Fig. 3a, lower left panel), as would be expected for a

random distribution. When implementing fluorophore-specific blinking, the normalized Ripley's K function resulted in a distinct peak indicating clusters of localizations. We next compared real protein clusters with their corresponding localization map as it manifested with fluorophore-specific blinking. Both scenarios feature clearly discernible clusters affecting Ripley's K function (Fig. 3a, lower right panel). Importantly, based on Ripley's K analysis the random distribution of blinking fluorophores (Fig. 3a, lower left panel) could not be discriminated from a clustered distribution of both non-blinking and blinking

**Fig. 3 Monte Carlo simulation-based approach to discriminate random from clustered molecular distributions of fluorophores featuring complex blinking patterns. a** Fluorophore blinking affects Ripley's K functions: Top: We simulated super-resolution images without blinking (gray) and with experimentally derived blinking parameters for PS-CFP2 (blue), mEOS3.2 (green), PA-JF549 (yellow), and CAGE635 (red) by distributing molecular detections randomly (left panel) or clustered (right panel, cluster radius: 60 nm, density of clusters: 20 cluster per $\mu m^2$ and 80% of molecules within clusters) at a density of 70 molecules per $\mu m^2$. Bottom: Ripley's K analysis of 15 simulations (mean ± SD) within an area of $15 \times 15\,\mu m^2$. Scale bars are 1 μm. **b** Comparison of Ripley's K functions derived from super-resolution images simulated under different clustering scenarios: cluster scenarios are classified as not resolvable, if mean values are within the respective confidence interval (top left), as borderline, if confidence intervals overlap at the maximum of Ripley's K function, but the mean values are not within the intervals, and as resolvable (top right), if confidence intervals do not overlap at the maximum of Ripley's K function (bottom left). Data are presented as mean values +/− SD. **c** Sensitivity of Ripley's K analysis in detecting clustering scenarios. Ripley's K functions were determined from Monte Carlo simulations of nanoclusters with a cluster radius of 20, 40, 60, or 100 nm, a fraction of molecules residing inside clusters between 20 and 100%, 3, 5, 10, 15, and 20 clusters per $\mu m^2$, an average molecular density of 70 molecules per $\mu m^2$ and the experimentally determined blinking statistics of PS-CFP2, mEOS3.2, PA-JF549, and CAGE635 determined under indicated imaging conditions. Functions were compared to those calculated from random localization distributions and categorized as described in **b**. **d** Classification of CD3ζ and LFA-1 distribution on fixed T cells: Top: Randomized distributions superimposed with experimentally determined blinking parameters of PS-CFP2 were simulated at densities of 86 (CD3ζ) and 145 (LFA-1) molecules $\mu m^{-2}$ (blue) for comparison with localization maps derived from PALM experiments involving CD3ζ-PS-CFP2 and anti-LFA-1-biotin visualized with mSAv*-cc-PS-CFP2* (magenta). Molecular densities were derived from experimental data by dividing the number of localizations by the average number of blinks. Bottom: Ripley's K functions of CD3ζ-PS-CFP2 and anti-LFA-1-biotin-mSAv*-cc-PS-CFP2* (magenta) were compared to Ripley's K functions derived from 10 simulations of randomized distributions with corresponding molecular densities (blue, mean ± SD). While the distribution of CD3ζ could not be discriminated against a random scenario, LFA-1 distribution deviated significantly. Scale bars are 1 μm. **e** Flow chart for cluster assessment.

fluorophores (Fig. 3a, lower right panel) without further knowledge of the fluorophores' blinking characteristics.

**Classifying Ripley's K functions to distinguish molecular clustering from random distributions.** Qualitative comparison of Ripley's K functions derived from randomized and clustered scenarios of blinking fluorophores (Fig. 3a, lower left and right, colored) revealed disparities, in particular with regard to their respective maximum values. Hence, with detailed knowledge of the blinking properties, a purely random distribution is quantitatively distinguishable from a clustered distribution through comparison of the respective Ripley's K function maxima (Fig. 3b). To assess the sensitivity of our combined approach, we simulated nanoclustering scenarios for each fluorophore in the presence of 4% PFA by varying the number of clusters, the number of molecules within clusters, and the cluster radius for a constant density of 70 molecules per square micron. We determined which scenarios were classified as not resolvable or borderline by means of Ripley's K function analysis and hence could be misinterpreted as random distributions (Fig. 3c). Only simulated clusters comprised of 20% of all recordable molecules or fewer evaded detection.

To assess whether the addition of 4% PFA changes the sensitivity in detecting protein clusters, we repeated simulations with fluorophore blinking data observed without the fixation agent (Supplementary Fig. 3E–H). While fewer clustering scenarios were resolvable when applying prolonged illumination times (Supplementary Fig. 4E–H), an even larger set of scenarios could in principle be resolved when imaging PS-CFP2 in the presence of GSH (Supplementary Fig. 5E).

For method validation we assessed the cell surface distribution of TCR-CD3ζ fused to PS-CFP2 (or mEOS3.2), or the TCR-CD3ζ chain decorated with the biotinylated H57 mAb, and the integrin LFA-1 decorated with a biotinylated LFA-1α chain-reactive antibody (mAb TS2/4) for detection via mSAv*-cc-PS-CFP2, mSAv*-cc-mEOS3.2, or mSAv*-c-CAGE635. As multiple biotins per mAb would result in the detection of artificial clusters, we kept the number of conjugated biotin molecules below two biotin molecules per mAb as confirmed by SDS-PAGE (Supplementary Fig. 6A, B).

T cells, either placed on ICAM-1- or on fibronectin-coated glass slides, were subjected to chemical fixation followed by PALM analysis applying the same experimental settings employed to determine the fluorescence protein's blinking properties (e.g., for PS-CFP2, 2 ms exposure at 3.0 kW cm$^{-2}$, PFA-treatment; refer to "Methods" section for details). Exposing cells to laser excitation without the addition of fluorophores yielded no or neglectable single-molecule detections (Supplementary Fig. 6C). As shown in Fig. 3d (left and right panel, red), localization maps of both CD3ζ-PS-CFP2 and LFA-1 (stained with the mAb TS2/4 and mSAv*-cc-PS-CFP2) featured clearly discernible clusters. However, comparison of the Ripley's K function resulting from the CD3ζ-PS-CFP2 PALM recording with those calculated from random distributions of simulated entities featuring PS-CFP2 blinking properties (Fig. 3d, left panel, blue) yielded no significant differences (Fig. 3d, left bottom panel), confirming previous results derived from a label density variation approach[39]. In contrast and consistent with earlier observations, antibody-labeled LFA-1 detected with mSAv*-cc-PS-CFP2 gave rise to localization clusters which could no longer be explained by overcounting only and which must hence have originated from LFA-1 nanoclustering[39,40] (Fig. 3d, right). Comparable results were obtained from PALM measurements of CD3ζ-mEOS3.2 and the TCRβ chain visualized with biotinylated H57 mAb and mSAv*-c-CAGE635 or antibody-labeled LFA-1 detected with mSAv*-cc-mEOS3.2 or mSAv*-c-CAGE635 (Supplementary Fig. 6D, E).

## Discussion

Unambiguous detection of one and the same single fluorescent dye revealed blinking behavior of photoswitchable and photoactivatable fluorophores in all its complexity under experimental conditions which were in essence identical to those employed in PALM imaging experiments. For proof of principle we first opted to quantitate the blinking of PS-CFP2, which had previously been described as less prone to blinking[27] and hence constituted an interesting candidate for SMLM. Our study shows that single PS-CFP2 molecules underwent several cycles of emissive states on timescales of several seconds, which are typical for super-resolution experiments. Only 26–50% of all PS-CFP2 molecules gave rise to exactly one detection, while the remaining 50–74% fluorophores were represented in the data set at least twice. For mEOS3.2, which is often employed in SMLM studies, we found a 1.6-fold increased average number of blinking events compared to PS-CFP2, with a fraction of 50–60% detected exactly once. We also included photoactivatable organic dyes in our comparison, which are at current rarely used but may with further

development make their way into PALM-based studies. While we observed the highest blinking tendency for CAGE635, the recently devised PA-JF549 outperformed PS-CFP2 with regard to (i) the fraction of molecules observed in one frame only (55% versus 35% at standard PALM conditions with 4% PFA) and (ii) shorter off-times. However, at the time of conducting this study, the absence of commercially available PA-JF549-maleimide or PA-JF549-biotin conjugates excluded site-specific conjugation to mSAv* for cell experiments. We also did not succeed in coupling commercially available PA-JF549-succinimidyl ester to 6-Methyl-Tetrazin-Amine for conjugation to mSAv*-TCO due to solubility issues in water-based buffers.

Our results also indicate that neither PFA-based fixation nor varying excitation power and illumination times affected the blinking behavior significantly (see Supplementary Table 1).

Of considerable importance for PALM analyses is that blinking of all fluorophores investigated here displayed distributions that did not follow a simple geometric function, unlike what has been previously described for single mEOS2 proteins[31,41]. Simple merging procedures, which are based on time stamping, fail to account for outliers with excessive blinking. As shown in this study, cluster analysis based solely on the use of normalized Ripley's K functions[38] or other cluster assessment tools cannot provide the means to distinguish between a random distribution of fluorophores, which blink like the fluorophores examined here, and a clustered non-random distribution of non-blinking molecules.

Of note, high-powered fluorophore excitation (at $15 \, kW \, cm^{-2}$) aimed to suppress PS-CFP2 blinking did altogether not solve the problem due to the resulting high number of unspecific signals which could not be distinguished from true PS-CFP2 detections. This was in large part because the brightness of unspecific background signals increased with rising excitation power density up to the level of fluorophore emission signals, with the latter remaining constant due to photon saturation. This finding highlights an additional advantage of the presented platform, which unlike cell-based PALM protocols informs on the fidelity of fluorophore detection at all chosen experimental settings.

It has previously been shown for the use of mEOS[31,41–44], PA-GFP, and PA-mCherry1[45] that detailed fluorophore characterization prior to conducting SMLM is instrumental for quantitative readout and maximally achievable resolution[46]. As is outlined in Fig. 3e, we arrived at the maximal sensitivity of PALM-based cluster detection by (i) precisely determining blinking statistics for given experimental conditions, (ii) comparing Ripley's K functions for simulated clustering scenarios, and through (iii) variation of cluster distribution parameters and subsequent classification. Hence, our SLB-based experimental platform may serve primarily the means to minimize blinking in PALM experiments through identification of suitable fluorophores and adequate imaging conditions, which would render simple merging and time stamping procedures sufficient to circumvent overcounting artifacts. If this is not possible as showcased here for PS-CFP2, mEOS3.2, PA-JF549, and CAGE635, we provide an alternative strategy involving quantitative Ripley's K analysis-based comparison of PALM-derived localization maps and of simulated event distributions derived from defined clustering scenarios and blinking properties.

Undercounting proteins due to non-functional fluorophores and other shortcomings in labeling is certain to affect the efficiency of cluster detection[47]. Our platform is suited to assess this risk as it provides the means to testing the functionality of purified fluorescent proteins and organic dyes via 2-color co-localization.

The detectability of clusters can be improved through rigorous testing of candidate fluorophores to be employed for imaging. As

was demonstrated in this study, the use of PS-CFP2 allowed for an unambiguous detection of more than 75% of all simulated clustering scenarios for PALM measurements under fixation conditions, which contrasts the detection of 62%, 60%, and 65% afforded by mEOS3.2, PA-JF549, and CAGE635, respectively. It also follows that the average number of blinking events measured for any given fluorophore is not sufficient for predicting its efficacy in cluster detection as the overall shape of the blinking parameter distribution plays a major role, too.

Of note, for best results, the presented classification via Ripley's K function can be readily substituted with other cluster assessment tools[23]. It may furthermore serve to improve the sensitivity of existing cluster assessment strategies such as fluorophore dilution[39], 2-color co-localization[48], and other measures to counteract artifacts resulting from fluorophore overcounting[49] and undercounting[47].

In summary, knowledge of how imaging conditions affect fluorophore blinking may well be the deciding factor when probing for subtle differences in clustering between individual samples. When aiming for meaningful two-color PALM-based co-localization, there is no viable alternative to identifying photoswitchable fluorophores featuring simple blinking behavior. The experimental approach described herein affords maximal control and critical fluorophore information to achieve this.

## Methods

**Animal model and ethical compliance statement.** 5c.c7 αβ TCR-transgenic mice bred onto the B10.A background were a kind gift from Björn Lillemeier (Salk Institute, USA). Animal husbandry, breeding, and sacrifice for murine T-cell isolation were evaluated by the ethics committees of the Medical University of Vienna and approved by the Federal Ministry of Science, Research and Economy, BMWFW (BMWFW-66.009/0378-WF/V/3b/2016). All animal-related procedures were performed in accordance with the Austrian law (Federal Ministry for Science and Research, Vienna, Austria), the guidelines of the ethics committee of the Medical University of Vienna, and the guidelines of the Federation of Laboratory Animal Science Associations (FELASA). Male and female mice 8–12 weeks of age were randomly selected for euthanasia, which was followed by the isolation of T cells from lymph nodes and spleen.

**Tissue culture.** T cells isolated from lymph nodes or spleen of 5c.c7 αβ TCR-transgenic mice were pulsed with 1 μM C18 reverse-phase HPLC–purified MCC 88-103 peptide (sequence: ANERADLIAYLKQATK, T-cell epitope underlined; Elim Biopharmaceuticals) and 50 U ml$^{-1}$ IL-2 (eBioscience) for 7 days[50] to arrive at an antigen-experienced T-cell culture. T cells were maintained at 37 °C in 1640 RPMI media (Life Technologies) supplemented with 10% FCS (Merck), 100 μg ml$^{-1}$ penicillin (Life Technologies), 100 μg ml$^{-1}$ streptomycin (Life Technologies), 2 mM L-glutamine (Life Technologies), 0.1 mM non-essential amino acids (Lonza), 1 mM sodium pyruvate (Life Technologies), and 50 μM β-mercaptoethanol (Life technologies) in an atmosphere containing 5% $CO_2$. T-cell experiments were conducted between day 7 and day 9 after isolation.

For retroviral production, we transfected the Phoenix packaging cell line with pIB2-CD3ζ-PS-CFP2 (or pIB2-CD3ζ-mEOS3.2) and the helper plasmid pcL-eco one day after T-cell isolation[37]. Phoenix cells were cultured at 37 °C and 5% $CO_2$ in DMEM (Life Technologies) supplemented with 10% FCS (Merck), 100 μg ml$^{-1}$ penicillin (Life Technologies), 100 μg ml$^{-1}$ streptomycin (Life Technologies), and 2 mM L-glutamine (Life Technologies). Three days after T-cell isolation, we subjected the cells to virus-containing supernatant from the Phoenix culture, 10 μg ml$^{-1}$ polybrene (Merck), and 50 U ml$^{-1}$ IL-2 (Merck) followed by spin infection for 90 min at 30 °C and $1000 \times g$. Addition of 10 μg ml$^{-1}$ blasticidin (Merck) 24 h after spin infection ensured selection of transduced cells. T cells expressing CD3ζ-mEOS3.2 or CD3ζ-PS-CFP2 were enriched by FACS (SH800, Sony) and used for experiments 1 day after cell sorting.

PBMCs isolated from human blood were thawed after storage in liquid nitrogen and cultured at 37 °C and 5% $CO_2$ in human T-cell medium, i.e., 1640 RPMI media (Life Technologies) supplemented with 10% human serum, 100 μg ml$^{-1}$ penicillin (Life Technologies), 100 μg ml$^{-1}$ streptomycin (Life Technologies), 2 mM L-glutamine (GlutaMAX™ Supplement, Gibco), and 50 μM β-mercaptoethanol (Life Technologies). To enrich human PBMCs for T cells, we seeded $1 \times 10^7$ PBMCs in a total volume of 3 ml into 6-well plates, which were pre-coated with 1 μg ml$^{-1}$ monoclonal antibody (mAb) OKT-3 (eBioscience) in 1× PBS. PBMCs were cultured in human T-cell medium supplemented with 100 U ml$^{-1}$ recombinant IL-2 (Novartis) and 0.5 μg ml$^{-1}$ mAb CD28.2 (eBioscience) for 48 h.

**Protein expression, refolding, and preparation of imaging tools.** The cysteine mutants of the monovalent streptavidin (mSAv* and mSAv*-3xHis$_6$) were prepared with some adaptions as described[51,52]. The pET21a (+) vectors encoding "alive" (i.e., biotin binding) and "dead" (i.e., biotin non-binding) streptavidin subunits were kindly provided by Alice Ting (Stanford University, USA). We substituted the 6x histidine tag of the "alive" subunit with a cleavable 6x glutamate tag to allow for purification via anion-exchange chromatography (MonoQ 5/50 GL) preceded by a recognition site of the 3C protease for optional removal of the tag (Supplementary Fig. 7). To arrive at mSAv*-3xHis$_6$, we extended the sequence of the "dead" subunit C-terminally with a 6x histidine tag (His$_6$) for attachment to lipid bilayers containing 18:1 DGS-NTA(Ni) (Supplementary Fig. 8). In the "alive" subunit, we substituted an alanine for a cysteine residue at position 106 (A106C) to produce a monovalent streptavidin that could be site-specifically conjugated to maleimide-linked fluorescent dyes (mSAv* or mSAv*-3xHis$_6$) (Supplementary Figs. 1 and 7). Both, "alive" and "dead" streptavidin subunits were expressed in *E. coli* (BL-21) for 4 h at 37 °C and refolded from inclusion bodies as described[52]. After refolding, the streptavidin tetramer mixture was concentrated in a stirred ultrafiltration cell (10 kDa cut-off, Merck). Further concentration and buffer exchange to 20 mM Tris-HCl pH 8.0 were carried out with Amicon Ultra-4 centrifugal filters (10 kDa cut-off, Merck). The mixture of tetramers was then purified by anion-exchange chromatography (MonoQ 5/50 GE Healthcare Life Sciences) using a column gradient from 0.1 to 0.4 M NaCl (Supplementary Fig. 1B). Monovalent streptavidin (mSAv* or mSAv*-3xHis$_6$) was eluted with 0.22 M NaCl, concentrated again (Amicon Ultra-4 centrifugal filters, 10 kDa cut-off) and further purified via gel filtration (Superdex-200 10/300 GE Healthcare Life Sciences). The protein mSAv*-3xHis$_6$ was either stored at −80 °C in 1× PBS or site-specifically labeled with Abberior STAR 635P maleimide (Abberior) or Alexa Fluor 488 C5 Maleimide (Thermo Fisher Scientific) according to the manufacturer's instructions and in the presence of 100 μM Tris(2-carboxyethyl)phosphine (TCEP, Thermo Fisher Scientific). To remove excess dye, (AF488- or) STAR635-conjugated mSAv*-3xHis$_6$ were purified by gel filtration using Superdex-75 (Superdex-75, 10/300 GL, GE Healthcare Life Sciences). Fractions containing monomeric (AF488- or) STAR635-conjugated mSAv*-3xHis$_6$ were concentrated with Amicon Ultra-4 centrifugal filters (10 kDa cut-off, Merck) to arrive at a protein concentration of ~1 mg ml$^{-1}$. The mSAv*-STAR635 or mSAv*-AF488 exhibited a protein to dye ratio of 1.0 as determined by spectrophotometry at 280/638 or 280/488 nm, respectively.

Monomeric photoswitchable cyan fluorescence protein 2 (PS-CFP2, Evrogen) and mEOS3.2[53] were C-terminally equipped with an AVI-tag (GLNDIFEAQKIEWHE) for site-specific biotinylation via the BirA ligase (Avidity) followed by a 3C protease cleavable (LEVLFQGP) 12x histidine (His$_{12}$) tag. The PS-CFP2-AVI-3C-His$_{12}$ construct was synthesized by Eurofins MWG Operon and shuttled into the pet21a (+) bacterial expression vector using the restriction enzymes NdeI and HindIII (Supplementary Fig. 9). To arrive at the mEOS3.2-AVI-3C-His$_{12}$ construct, we replaced PS-CFP2 for mEOS3.2 with the restriction enzymes NdeI and BamHI (Supplementary Fig. 10). Optionally, we substituted a serine for an unpaired cysteine in the linker sequence of PS-CFP2 between the AVI-tag and the 3C protease cleavage site (see Supplementary Fig. 9) to create a target site for site-specific conjugation via click chemistry (PS-CFP2*); mEOS3.2 has an unpaired cysteine that can be functionalized with maleimide chemistry. All constructs, PS-CFP2-AVI-3C-His$_{12}$, PS-CFP2*-AVI-3C-His$_{12}$, and mEOS3.2-AVI-3C-His$_{12}$ were expressed as soluble proteins in *E. coli* (BL-21) for 3 h at 30 °C. Bacterial cells were processed via ultrasound, subjected to ice-cold nickel binding buffer containing 50 mM Tris-HCl pH 8.0, 300 mM NaCl (Merck), 10 mM imidazole (Merck), 1 mM PMSF (Merck), and a protease inhibitor cocktail (complete™, Roche), and subjected to centrifugation at 10,000 × *g*. The supernatant was filtered (Filtropur S 0.2 μm, Sarstedt) and subjected to Ni-NTA based affinity chromatography (QIAGEN). Eluted protein was further purified via anion-exchange chromatography (MonoQ 5/50 GE Healthcare Life Sciences) and gel filtration (Superdex-200 10/300 GE Healthcare Life Sciences). Fractions containing monomeric PS-CFP2 (or mEOS3.2) were concentrated with Amicon Ultra-4 centrifugal filters (10 kDa cut-off, Merck) and the maturation state of the fluorescent PS-CFP2 or mEOS3.2 protein was verified via protein absorbance at 280/400 or 280/507 nm, respectively. Monomeric PS-CFP2 (or mEOS3.2) was site-specifically biotinylated using the BirA ligase (Avidity) to obtain the molecular probe PS-CFP2-biotin (or mEOS3.2-biotin). The 12× histidine tag was removed by overnight digestion with 3C protease (GE Healthcare Life Sciences) followed by a purification step with Ni-NTA agarose (QIAGEN) to separate the cleaved PS-CFP2-biotin (or mEOS3.2-biotin) from 3C protease (His-tagged) and unprocessed PS-CFP2-biotin-His$_{12}$ (or mEOS3.2-biotin-His$_{12}$). The unbound protein fraction containing PS-CFP2-biotin (or mEOS3.2-biotin) was again purified via gel filtration (Superdex-200 10/300 GE Healthcare Life Sciences). Monomeric PS-CFP2-biotin (or mEOS3.2-biotin) was snap-frozen in liquid N$_2$ and stored in 1× PBS at −80 °C.

After refolding and purification, fractions containing monomeric PS-CFP2*-AVI-3C-His$_{12}$ (or mEOS3.2-AVI-3C-His$_{12}$) were concentrated using Amicon Ultra-4 centrifugal filters (10 kDa cut-off, Merck) in the presence of 100 μM Tris(2-carboxyethyl)phosphine (TCEP, Thermo Fisher Scientific). Monomeric PS-CFP2*-AVI-3C-His$_{12}$ (or mEOS3.2-AVI-3C-His$_{12}$) was site-specifically conjugated with 6-Methyl-Tetrazine-PEG4-Maleimide (Jena Bioscience) to arrive at PS-CFP2*-tetrazine (or mEOS3.2-tetrazine). The conjugates were purified via S75 gel filtration (Superdex-75, 10/300 GL, GE Healthcare Life Sciences) and further processed as described below.

To arrive at a conjugate of mSAv*-STAR635 and PS-CFP2-biotin (or mEOS3.2-biotin), we incubated both probes with a 12-fold excess of PS-CFP2-biotin (or mEOS3.2-biotin) at high concentrations (>1 mg ml$^{-1}$) for 2 h at RT in 1 x PBS and purified the resulting construct mSAv*-STAR635:PS-CFP2-biotin (or mSAv*-STAR635:mEOS3.2-biotin), which can bind to DGS-NTA(Ni) on SLBs, together with PS-CFP2-biotin (or mEOS3.2-biotin), which cannot bind to DGS-NTA(Ni) on SLBs, via gel filtration (Superdex-200 10/30, GE Healthcare Life Sciences). Resulting peak fractions containing only the mSAv*-STAR635:PS-CFP2-biotin or mSAv*-STAR635:mEOS3.2-biotin (as verified via SDS-PAGE, Supplementary Fig. 1C, D) were stored in 1× PBS and 50% glycerol at −20 °C.

To produce the conjugate mSAv*-AF488 and CAGE635-biotin (Abberior), we incubated both reactants in a 1:40 molar ratio at high concentrations (>1 mg ml$^{-1}$) for 2 h at RT in 1× PBS and purified the resulting construct mSAv*-AF488:CAGE635-biotin as described above.

To generate the conjugate mSAv*-STAR635 and PA-JF549, we initially coupled PA-JF549 Succinimidyl Ester (Tocris) to biocytin (Thermo Fisher Scientific) in a 3 to 1 molar ratio in a 90/10 DMSO/water solution and purified the resulting adduct PA-JF549-biocytin via C18 reversed-phase HPLC (1260 Infinity II, Agilent Technologies). mSAv*-STAR635 was then incubated with PA-JF549-biocytin at high concentrations (>1 mg ml$^{-1}$) for 2 h at RT in 1× PBS and purified as described above to arrive at mSAv*-STAR635:PA-JF549-biocytin.

To combine PS-CFP2* (or mEOS3.2*) with monovalent streptavidin (mSAv*) while keeping the biotin-binding site unoccupied, we site-specifically conjugated PS-CFP2* (or mEOS3.2) with 6-Methyl-Tetrazine-PEG4-Maleimide (Jena Bioscience) and the monovalent streptavidin (mSAv*) with TCO-PEG3-Maleimide (Jena Bioscience) at their free cysteines according to the manufacturer's instructions. PS-CFP2*-tetrazine (or mEOS3.2-tetrazine) and mSAv*-TCO were purified from free 6-Methyl-Tetrazine-PEG4-Maleimide and TCO-PEG3-Maleimide, respectively, via gel filtration (Superdex-75 10/300; GE Healthcare Life Sciences). We incubated PS-CFP2*-tetrazine (or mEOS3.2-tetrazine) with mSAv*-TCO in a 5:1 molar ratio at high concentrations (>1 mg ml$^{-1}$) for 1 day at 4 °C in 1× PBS. The resulting adduct mSAv*-cc-PS-CFP2* (or mSAv*-cc-mEOS3.2) was purified together with an excess of PS-CFP2*-tetrazine (or mEOS3.2-tetrazine) via Ni-NTA-based affinity chromatography (QIAGEN) and gel filtration (Superdex-200 10/30, GE Healthcare Life Sciences). Fractions containing mSAv*-cc-PS-CFP2* (or mSAv*-cc-mEOS3.2), which can bind to biotin, as well as unreacted PS-CFP2*-tetrazine (or mEOS3.2-tetrazine), which cannot bind to biotin, were identified via SDS-PAGE, concentrated with Amicon Ultra-4 centrifugal filters (10 kDa cut-off, Merck) and stored in 1× PBS supplemented with 50% glycerol at −20 °C.

To produce mSAv*-c-CAGE635, we conjugated mSAv* to Abberior CAGE635 Maleimide (Abberior) according to the manufacturer's instructions and purified and stored the probe as described above.

The human CD11a (LFA-1α chain) reactive mAb TS2/4 (BioLegend) and the murine TCRβ chain-reactive mAb H57 (BioLegend) were conjugated to EZ-Link™ NHS-LC-LC-Biotin (Thermo Fisher Scientific) in a 1 to 2 molar ratio in the presence of 0.1 M NaHCO$_3$ for 1 h. Biotinylated mAbs were purified from non-biotinylated mAbs and unconjugated EZ-Link™ NHS-LC-LC-Biotin via monomeric avidin-based affinity chromatography (Immobilized Monomeric Avidin, Thermo Fisher Scientific). Biotinylated mAbs were eluted from the avidin agarose column with 2 mM biotin and further purified via gel filtration (Superdex-200 10/300, GE Healthcare Life Sciences), concentrated using Amicon Ultra-4 centrifugal filters (10 kDa cut-off, Merck) and stored in 1× PBS at 4 °C. The degree of mAb TS2/4 or mAb H57 biotinylation was verified by a monovalent streptavidin-based gel shift assays and SDS-PAGE analysis (Supplementary Fig. 6A, B).

ICAM-1-His$_{12}$ was cloned into the pAcGP67 vector (BaculoGold™ Baculovirus Expression System, BD Biosciences) for baculovirus production as described[54]. To produce ICAM-1-His$_{12}$, we infected High Five™ cells (BTI-TN-5B1-4, Thermo Fisher Scientific) with the baculovirus according to the manufacturer's instructions (BD Biosciences). The supernatant was harvested 3–4 days after infection at 90% cell viability by centrifugation and filtration (0.45 μm) and dialyzed against 1× PBS by tangential flow filtration (Minimate™ TFF System equipped with 10 kDa T-Series cassettes, Pall Corporation). ICAM-1-His$_{12}$ was then subjected to Ni-NTA-based affinity chromatography (HisTrap GE Healthcare), anion-exchange chromatography (MonoQ 5/50 GL, GE Healthcare), and gel filtration (Superdex-200 10/300 GL, GE Healthcare). The protein was concentrated with Amicon Ultra-4 centrifugal filters (10 kDa cut-off, Merck) and stored in 50% glycerol at −20 °C.

**SDS-PAGE and protein band visualization.** Samples were mixed with a 4× loading buffer (252 mM Tris-HCl, 40% glycerol, 8% SDS, and 0.04% bromophenol blue, pH 6.8) with or without 20 mM dithiothreitol (Merck) for reducing and non-reducing conditions, respectively, and then subjected to 10% SDS-PAGE in running buffer (25 mM Tris-HCl, 192 mM glycine and 0.1% SDS, pH 8.2) followed by colloidal Coomassie Brilliant Blue G-250[55] staining or silver staining[56].

**Preparation of glass-supported lipid bilayers and fibronectin-coated surfaces.** One-hundred micrograms of 1,2-dipalmitoyl-*sn*-glycero-3-phosphocoline (DPPC) and 0.75 μg of 1,2-dioleoyl-*sn*-glycero-3-[*N*(5-amino-1-carboxypentyl)iminodiacetic acid] succinyl[nickel salt] (DGS-NTA(Ni); Avanti Polar Lipids) were mixed in chloroform, dried under N$_2$, dissolved in 1 ml PBS at 50 °C, and bath-sonicated at 50 °C. Glass cover slips (#1.5, 24 × 60 mm, Menzel) were plasma-cleaned for 10 min

in an Harrick Plasma Cleaner (Harrick) and attached to 8-well LabTek chambers (Nunc), for which the bottom glass slide had been removed, with two-component dental imprint silicon glue (Picodent twinsil 22, Picodent). Slides were incubated with the vesicle suspension in PBS for 15 min and afterward rinsed with 15 ml 1× PBS. Histidine-tagged protein complexes (e.g., mSAv*-STAR635:PS-CFP2-biotin) were added in concentrations yielding a single-molecule density, incubated for 75 min at 25 °C and rinsed with 15 ml 1× PBS. For experiments involving paraformaldehyde-based fixation, the samples were fixed with 4% paraformaldehyde (PFA; Polysciences) in 1× PBS for 10 min at 25 °C and subsequently rinsed with 1× PBS.

For T-cell experiments, we prepared murine ICAM-1 and fibronectin-coated surfaces. For this, we incubated plasma-cleaned glass cover slips (glued to 8-well LabTek chambers) with 10 µg ml⁻¹ ICAM-1-His₁₂ or 30 µg ml⁻¹ fibronectin (Merck) in 1 x PBS for 2 h at 37 °C. Chambers were washed in imaging buffer (HBSS, Gibco, supplemented with 1% FCS, 2 mM MgCl₂, and 2 mM CaCl₂) before addition of cells.

**Preparation of cell samples for microscopy**. The samples for CD3ζ-PS-CFP2 experiments were prepared as previously described[37]. We followed the same procedures for CD3ζ-mEOS3.2 experiments. In brief, 1 × 10⁶ CD3ζ-PS-CFP2 (or CD3ζ-mEOS3.2) expressing 5c.c7 αβ TCR-transgenic T cells were washed once into imaging buffer (HBSS supplemented with 1% FCS, 2 mM MgCl₂, and 2 mM CaCl₂) and seeded onto ICAM-1-His₁₂-coated glass slides. After cell attachment for 15 min, samples were carefully washed with 1× PBS and fixed with 4% PFA in 1× PBS for 25 min at 4 °C. Fixation was stopped by washing the sample with imaging buffer (HBSS supplemented with 1% FCS, 2 mM MgCl₂, and 2 mM CaCl₂). Samples were carefully washed into 1× PBS prior to imaging.

For TCRβ chain PALM measurements, antigen-experienced 5c.c7 αβ TCR-transgenic T cells were incubated with 20 µg ml⁻¹ CD16/32 mAb (clone 93, BioLegend) and subsequently stained with 40 µg ml⁻¹ H57-biotin mAb for 30 min on ice followed by a washing step in 15 ml imaging buffer (HBSS supplemented with 1% FCS, 2 mM MgCl₂, and 2 mM CaCl₂) and a staining step employing 50 µg ml⁻¹ mSAv*-c-CAGE635 for 30 min on ice. After washing, T cells were allowed to adhere to ICAM-1-His₁₂ coated glass slides for 15 min before fixation with 4% PFA as described above.

For CD11a PALM measurements, 0.3 × 10⁶ human PBMC-derived T cells were incubated with 3% beriglobin (CSL Behring) for 10 min and subsequently stained with 40 µg ml⁻¹ mAb TS2/4-biotin for 30 min on ice followed by a washing step in 15 ml imaging buffer (HBSS supplemented with 1% FCS, 2 mM MgCl₂, and 2 mM CaCl₂) and a staining step employing 50 µg ml⁻¹ mSAv*-cc-PS-CFP2 (or mSAv*-cc-mEOS3.2, or mSAv*-c-CAGE635) for 30 min on ice. After an additional washing step, cells were allowed to spread on fibronectin-coated glass slides for 15 min at 25 °C before fixation with 4% PFA as described above.

**Microscopy**. Objective-based TIRF microscopy was conducted using an inverted microscope (Axiovert 200, Zeiss) equipped with a chromatically corrected objective (100× N.A. = 1.46, Plan-Apochromat, Zeiss) and 405 nm (photoswitching of PS-CFP2 and mEOS3.2 and photoactivation of CAGE635 and PA-JF549; iBeam smart, Toptica), 488 nm (imaging of AF488 and PS-CFP2; optically pumped semiconductor, Sapphire; Coherent), 532 nm (imaging of mEOS3.2 and PA-JF549; LCX532L with additional AOM, installed in L6Cc laser combiner, Oxxius), and 647 nm (imaging of STAR635 and CAGE635; iBeam smart, Toptica) laser sources. For rapid shuttering of the 488 nm laser illumination, an acousto-optical modulator was used (1205C, Isomet). Fluorescence emission was detected with a back-illuminated EMCCD camera (iXon Ultra 897, Andor. Timing protocols were generated by in-house developed programs implemented in Labview (National Instruments) and executed by a built-in digital/analog IO-card (model PCI-6713, National Instruments).

Single mSAv*-STAR635 (or mSAv*-AF488) molecules were tracked using 647 nm (or 488 nm) excitation with a 100 ms time lag for at least 50 frames and employing a power density of 1.5 kW cm⁻² during the illumination time of 2 ms.

For determination of the mobility over longer time periods, tracking sequences were acquired with the mSAv*-STAR635 probe using 647 nm excitation with a time lag of 500 ms over 120 frames; power density (1.5 kW cm⁻²) and illumination time (two ms) remained unchanged. For photoswitching or photoactivation, the 405 nm laser was operated at continuous-wave illumination at a power density of 20–30 W cm⁻².

For PS-CFP2 PALM measurements, 2 and 15 ms of illumination at ~3 and ~0.4 kW cm⁻², respectively, were followed by a delay time of 4 ms which was necessary to read out the cropped camera chip yielding frames rates of 167 Hz (2 ms illumination) and 53 Hz (15 ms illumination). A total of 10,000 frames were recorded. For CAGE635 PALM measurements, we recorded 10,000 frames with 2 and 15 ms of illumination at ~4 kW cm⁻²/167 Hz frame rate and ~0.53 kW cm⁻²/53 Hz frame rate, respectively, and a delay time of 4 ms. Excitation light was uncoupled from emission light with the use of a dichroic mirror (zt488/640rpc; Chroma). Emission was then was split by a Dual View system (Photometrics) equipped with a 640dcxr dichroic mirror and HQ700/75 (both Chroma) and 525/45 (Semrock) emission filters.

For mEOS3.2 and PA-JF549 PALM measurements, we recorded 10,000 frames with a delay time of 4 ms, and 2 and 15 ms of illumination at ~4 kW cm⁻²/167 Hz

frame rate and ~0.57 kW cm⁻²/53 Hz frame rate, respectively. Excitation light was uncoupled from emission light with the use of a dichroic mirror (zt405/488/532/640rpc; Chroma). Emission was then was split by a Dual View system (Photometrics) equipped with a 640dcxr dichroic mirror and HQ700/75 and HQ585/40 m (all three filters Chroma) emission filters. For control measurements on cells and SLBs without any fluorescent label, an ET605/52 filter (Chroma) was used instead of the HQ585/40 m mentioned above.

All microscopy experiments involving T cells were performed employing the same microscopy setup, illumination schemes, and laser powers used for recording the respective platform data.

**Single-molecule blinking analysis**. Single molecules appearing in both channels were detected and localized by using a Maximum Likelihood Estimator implemented in the ThunderSTORM ImageJ plugin[57].

To determine the mobility of mSAv*-STAR635 on the DPPC bilayer, a published[58] algorithm implemented in MATLAB was used for the generation of trajectories, which were subjected to a mean square displacement analysis.

Independently of tracking, the position of mSAv*-STAR635 (or mSAv*-AF488) molecules was averaged during the imaging period by using the localization merging algorithm implemented in ThunderSTORM with the following parameters: maximum $t_{off}$ = 10,000 frames, maximum displacement = 1 pixel. This mean position was then employed for co-localization analysis.

Determined positions of mSAv*-STAR635 (or mSAv*-AF488) were corrected for chromatic aberration by an affine transformation matrix, which was experimentally derived from imaging TetraSPECK beads (Thermo Fisher Scientific). mSAv*-STAR635 (or mSAv*-AF488) molecules with a nearest neighbor within a distance smaller than 500 nm were discarded.

Fluorescent signals from individual labels (i.e., PS-CFP2, mEOS3.2, PA-JF549, or CAGE635 molecules) were grouped via hierarchical agglomerative clustering using the Euclidean distance metric. Unweighted average distance (UPGMA) was selected as linkage criterion. The resulting dendrogram was cut at 200 nm to obtain individual clusters. Localization clusters were regarded as colocalized if a platform signal was located within a radius of 500 nm from a cluster center. Only colocalized localization clusters were selected for further analysis. A localization cluster threshold (see software documentation) was set to 100 (for mEOS3.2 15 ms/0.57 kW/cm⁻² and PA-JF549 15 ms/0.53 kW/cm⁻²), 150 (mEOS3.2 2 ms/4 kW/cm⁻² and 2 ms/4 kW/cm⁻² + 4% PFA, and PA-JF549 2 ms/4 kW/cm⁻² + 4% PFA), and 600 (for CAGE635 15 ms/0.53 kW/cm⁻²) to filter out extreme outliers. Less than three fluorophore signals were lost for every filter applied.

For plotting the normalized cumulative sum of detections, a vector containing the number of detections for all molecules analyzed (i.e., the variable blink_dist. num of provided data) was first sorted in ascending order and the cumulative sum of detections was calculated. The new vector was finally normalized by division with the overall number of detections and multiplication with the average number of detections for a particular molecule.

**Simulations**. A 15 × 15 µm² region of interest featuring molecules at specified densities was simulated as described[37]. Briefly, a probability mask was generated by placing centers of clusters randomly according to a uniform distribution, and distributing the positions of molecules within clusters based on a two-dimensional Gaussian distribution located at the cluster center and truncated at the cluster size, i.e., one time the standard deviation of the Gaussian. A given proportion of molecules was attributed to clusters (% of molecules in cluster). Remaining molecular positions were randomly added on areas outside the clusters. To include blinking, the number of detections per label was drawn from the experimentally derived probability distribution of $N$. Finally, positions were shifted into a random direction by a distance drawn from a normal distribution with mean 0 and experimentally derived localization precision as standard deviation. For performing simulations used for comparison with cell-associated microscopy data, the approximate expression levels of CD3ζ-PS-CFP2, CD3ζ-mEOS3.2, TCRβ (stained with the mAb H57-biotin and fluorophore-conjugated mSAv*), and LFA-1 (stained with the mAb TS2/4-biotin and fluorophore-conjugated mSAv*) within a region of interest was determined by dividing the number of localizations with the mean number of detections per fluorescence molecule.

**Reporting summary**. Further information on research design is available in the Nature Research Reporting Summary linked to this article.

## Data availability
Source data are provided with this paper. The data that support the findings of this study are available through "figshare.com" with the identifier(s) https://doi.org/10.6084/m9.figshare.12871538[59]. The datasets generated during and/or analyzed during the current study are available from the corresponding authors upon reasonable request. Source data are provided with this paper.

## Code availability
The complete software package for channel registration, the analysis of blinking behavior and the comparison of cell-associated microscopy data with simulations is available for

download via Github and Zenodo with the identifier(s) https://doi.org/10.5281/zenodo.4003734[60].

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

## Acknowledgements

This work was supported by the Austrian Science Fund (FWF) through the PhD program Cell Communication in Health and Disease W1205 (J.B.H. and H.S.), the projects I953-B20, I4662-B (M.B.), V538-B26 (E.S.), P27941-B28 (F.B.), P26337-B21, P25730-B21, P30214-N36, F6809N36 (G.J.S.), P25775-B2 (J.B.H.), and the Vienna Science and Technology Fund (WWTF) projects LS13-030 (G.J.S and J.B.H.) and LS14-031 (J.B.H.). Funding was further provided by a predoctoral fellowship from the Boehringer Ingelheim Fonds (R.P.).

## Author contributions

R.P., G.J.S., J.B.H., and M.B. conceived the project. B.K.R., R.P., M.C.S., G.J.S., J.B.H., and M.B. wrote the manuscript. R.P. developed the linker system, contributed all probes, and performed imaging experiments. M.B. analyzed data. B.K.R. performed imaging experiments, analyzed data. B.K.R. and M.C.S. developed software tools. E.S., F.B., and H.S. contributed important ideas.

## Competing interests

The authors declare no competing interests.
