## [Peer Review File · Nature Communications]

Reviewers' comments:

Reviewer #1 (Remarks to the Author):

Rosboth et al. propose in their manuscript "Unscrambling Fluorophore Blinking for Comprehensive Cluster Detection via PALM" a method to overcome clustering artifacts caused by overcounting of blinking single molecules in PALM super-resolution microscopy. The new method offers an alternative to established "merging and time stamping" procedures where molecule detection events are combined into single events when they happen in a short enough time window. Importantly, the new method works also when blinking events are separated over longer stretches of time. The authors show, that this scenario is the case for the photo-switchable fluorescent protein PS-CFP2. In the manuscript, they characterize the observed blinking behavior of PS-CFP2, develop a method where clustering analyses from Ripley's K function are compared between observed protein distributions and simulations based on known spatial distributions and the previously obtained blinking behavior, and test it at the example of T-cell plasma-membrane located CD3zeta and integrin LFA-1.

Detecting clustering reliably using super-resolution microscopy is of importance, especially in the field of plasma membrane located receptor signaling. The manuscript has however a number of major weaknesses which, in my opinion, need to be addressed before publication.

- 1) The manuscript could benefit from more careful and clearer descriptions and definitions. The reader is often left guessing what the authors exactly meant or did or has to hunt through the manuscript to find the exact definition of a used term.
- 2) PS-CFP2 is, compared to Eos or organic STORM dyes, very dim. This can make identifying a molecule in a given frame very unreliable since detection threshold or background levels have a major influence on the detection efficiency. In the worst case, even small changes in background levels or threshold in the analysis algorithm can cause strong changes in the perceived blinking characteristics of PS-CFP2 (e.g. leading to completely different numbers of "gaps" or "detections"). This effect potentially renders the demonstrated simulation approach very unreliable. The authors should systematically investigate the robustness of their approach, for example how the data presented in Fig. 1E changes when data analysis parameters such as detection thresholds or experimental conditions (e.g. background or signal levels) change. The current statement "However, background fluorescence increased under these circumstances to levels which no longer allowed for a clear discrimination between localizations of single PS-CFP2 molecules and of background signals (Supplementary Fig. 2B)." (line 133) is in my view not sufficient.
- 3) Software code plays a major role in this manuscript. A major selling point of the manuscript is, that the method is readily applicable by the community. The authors mention that their code is available for download, but do not provide any details where and how. Similarly, no documentation of the code or a manual are provided. This should be added if the developed tool is supposed to have an impact.
- 4) Environmental conditions can influence the blinking behavior of fluorescent probes (as the authors state themselves for glutathione). Yet, for their T-cell experiments, they fix the cells, which was not done in the in vitro experiments, if I am not mistaken. It is unclear how this treatment compares to the experiments done with purified proteins which are used as the base of the simulations. I suggest that the authors characterize PS-CFP2 under fixation conditions and use this data for the corresponding simulations.
- 5) Page 10, Lines 206-209: The authors describe the observation of clusters when imaging antibody-labeled LFA-1 and explain this with clustering of LFA-1. Another possibility would be that multiple biotins are bound to each antibody. How did the authors rule out this possibility?
- 6) A discussion of possible artifacts arising from "undercounting" (by not labeling every molecule or bleaching individual fluorescent proteins before they could be detected, for example) is completely missing. The authors should briefly explain why undercounting does not affect their analysis or how they addressed this issue.
- 7) The discussion of the data summarized in Fig. 1E needs to include a statistical analysis to clarify if any of the observed changes are statistically significant.

Additional minor comments:

- Line 103: "When applying an excitation power density of 3.0 kW/cm² for 2 ms, a single PS-CFP2 molecule was detected on average 3.47 times over the course of 10,000 frames" – what does the "2 ms" mean here? It should not be necessary to extract this kind of simple information from the methods section. Similarly, on page 6, line 104, it would be helpful to mention the wavelength and frame rate.
- Given the central importance of the term "detection" in this manuscript, it should be clearly and explicitly defined, which it currently is not. I interpret the different mentions (in particular, line 98 combined with Fig. 1D) to mean that a "detection" for the authors is the appearance of a molecule in one frame, and not that multiple occurrences of a molecule in one emission burst are merged into one "detection".
- What is a "PS-CFP2 signal" (line 103)? It seems to be the trace of a single molecule since the number 1080 appears in the context of "signal" as well as the number of molecules in Fig. 1E. It would be helpful to not have the reader speculate about this by providing a clear definition and by using consistent terminology throughout the manuscript.
- On page 3, I suggest to include mMaple in the list of photoswitchable fluorescent proteins.
- Page 5, lines 85-86: the authors call signals which only show green signal but no STAR635 signal "false positive". This terminology suggests that these signals represent "trace amounts of (partly) hydrophobic buffer- or lipid-derived dyes" (lines 74-75). It is, however, from this reviewer's perspective equally possible that STAR635 labeling efficiency of mSav* was far from 100% which would explain the large number of green-only signals as well. The authors should discuss this possibility in their manuscript and either consider it in their later reasoning or clarify why they can rule out this possibility.
- The description of how Fig. 2A was generated (bottom of page 7) is very cryptic. I believe I have figured it out after quite some thought, but I am still not 100% sure that I got it. This can probably be fixed by more careful phrasing and consistent use of definitions. For example, what does "sorting the detections per molecule in ascending order" (line 145) mean? Do the authors mean "sorting the molecules by their number of detections"? Are "fluorophore localizations" (line 146) the same as "detections"? Overall, it is not clear what information Fig. 2A provides that is not already provided in Fig. 1E.
- First paragraph on page 8: Why are the maxima of the plots in Fig. 2A not exactly equal to the mean number of detections per molecule?
- What are the "blinks" mentioned in Fig. 2B? Are "detections" in subsequent frames combined into one emission event (as it is usually done), or not? This needs to be declared at least, or better, both images (with and without the combination of subsequent detections into one event) should be displayed.
- Given the importance of the confidence intervals for Ripley's K function in the developed method, it would be helpful to explicitly describe how they are obtained. It seems that it is the S.D. of 15 simulations, but for what field of view and what number of molecules? One can collect it from little pieces of information throughout the manuscript, but a clear statement where the confidence intervals are first introduced would lower the frustration of readers and avoid misunderstandings.
- Fig. 2D: why is the number of clusters per μm^2 used as a relevant parameter? It seems to me that the total number of molecules would be a more relevant parameter, since the number of clusters per μm^2 makes only sense for a particular field of view and therefore makes it difficult to compare between microscopes.

Reviewer #2 (Remarks to the Author):

The manuscript from Rossboth et al. addresses the issue of molecular blinking of PSCFP2 and its influence in proper molecular counting and cluster detection in the context of PhotoActivated Localization Microscopy. The authors perform most of their experiments in-vitro, and provide in the end an application in fixed T cells. The findings are then generalized to propose a pipeline to use blinking information to sort in a rigorous way clustering estimator (such as Ripley's K/L

functions) to determine to what extent a sample is displaying true oligomerization or artifactual clustering due to repeated counting of the same molecule.

I have several major concerns with this manuscript.

The first and foremost, is that the authors address a question that has been around since about a decade now (see Lee et al. PNAS 2012, Annibale et al JPCCL 2010), focusing on a first-generation fluorescent protein, PS-CFP2, which has also been used since 2007. Since then, a large number of reports (most recently, and to cite only a few, Fricke et al. Sci Rep 2017, Rollins et al. PNAS 2014, Dursic et al. Nat. Methods 2014) have addressed this issue proposing a wide array of methods to characterize photoblinking/single molecule photophysics and include it in the resulting image analysis process in order to avoid counting/clustering artifacts. Given the crowded field, it is not clear what are (i) the novelty and (ii) the generality of the findings and of the approach proposed by the authors.

While one may argue that PS-CFP2 has so far been considered a largely non-blinking fluorophore and any information on its true blinking behavior should be disseminated, nevertheless, the current manuscript unfortunately does not focus on providing an exhaustive spectroscopic characterization of the behavior of PS-CFP2. The photophysical characterization of the fluorophore behavior in the current manuscript falls short of the state of the art. When the reversible photo switching behavior of Dronpa, a reversible photoswitching fluorescent protein, was first investigated (Habuchi et al. PNAS 2005), the authors measured Dronpa on a confocal setup, using Avalanche Photodiodes: this yields the sub ms-temporal resolution necessary to obtain a credible photophysical model for the behavior of the fluorophore. A number of questions about CFP2 blinking thus remain unanswered:

What is the role of pH in determining the blinking behavior of PS-CFP2? Furthermore, only two power levels (a third is deemed to generate too high background) are used to characterize the change of the blinking parameters. Is this sufficient to extract a meaningful photophysical model for PSCFP2, as done in Fig. 5 of Habuchi et al? Can different combinations of photoactivation (405 nm) and excitation (488 nm) light contribute to modulate the blinking pattern? What about the behavior of the fluorophore in the Cyan, non-photoconverted form?

Finally, it is not clear how the pipeline which the authors provide, namely to categorize Ripley's function by taking into account the 'blinking fingerprint' of the fluorophores, can be generalized to other fluorescent proteins, and possibly, to other dyes, such as those used in STORM. Since 2006, and the development of PALM and STORM, the two techniques have become somehow interchangeable: STORM being favored for the higher brightness of the organic dyes used and the typically higher resolution of the reconstructed sub-diffraction limited structures. However, given the stochastic, rapid blinking of STORM dyes, the possibility to count molecules in STORM (see Finan et al. Angewandte 2015) is much more challenging than with fluorescent proteins. For any claim of generality, my recommendation to the authors would be to definitely expand their investigation to other FPs and also organic dyes.

Other Major comments:

The authors in Fig. 1 provide two representative traces, as well as statistics based on approximately 1000 molecules. Given the sophisticated single molecule imaging platform which they use, one would expect it would be possible to harvest a significant larger amount of data. Furthermore, if the point is that outlier's behavior does matter, then a 1% outlier population is represented here by only about 10 molecules, which is a rather limited statistical sample.

The authors propose a 'comprehensive methodology' to discriminate datasets based on comparing Ripley's functions, as done in Fig. 2c-d. I.e., generating Ripley's functions of spatially random datasets, affected by the blinking behavior. Previous work has however been done along these lines (Shivanandan et al, PlosOne 2015). In addition, one would expect that Ripley's functions' behavior to be affected also by factors like the localization error of the fluorophore, as well as the

total number of fluorophores sampled, i.e. the local sample density, therefore unless these cases are explicitly discussed, it becomes unclear to determine how quantitatively reliable the comparison of Ripley's functions of heterogeneous samples to simulated random controls can be in assessing their effective degree of clustering.

AUTHORS' POINT-BY-POINT RESPONSE TO THE REVIEWERS' COMMENTS

The reviewers' comments are written in Arial Narrow font type. Our responses are introduced with in arrow (→) and set in brick red Times New Roman.

Reviewer #1:

Rosboth et al. propose in their manuscript "Unscrambling Fluorophore Blinking for Comprehensive Cluster Detection via PALM" a method to overcome clustering artifacts caused by overcounting of blinking single molecules in PALM super-resolution microscopy. The new method offers an alternative to established "merging and time stamping" procedures where molecule detection events are combined into single events when they happen in a short enough time window. Importantly, the new method works also when blinking events are separated over longer stretches of time. The authors show, that this scenario is the case for the photo-switchable fluorescent protein PS-CFP2. In the manuscript, they characterize the observed blinking behavior of PS-CFP2, develop a method where clustering analyses from Ripley's K function are compared between observed protein distributions and simulations based on known spatial distributions and the previously obtained blinking behavior, and test it at the example of T-cell plasma-membrane located CD3zeta and integrin LFA-1.

Detecting clustering reliably using super-resolution microscopy is of importance, especially in the field of plasma membrane located receptor signaling. The manuscript has however a number of major weaknesses which, in my opinion, need to be addressed before publication.

→ We thank Reviewer #1 for his/her time and efforts to study our work in much detail and to carefully assess its integrity. After 9 months of intensive work efforts we now feel confident that we have addressed all issues he/she has raised (see below).

1) The manuscript could benefit from more careful and clearer descriptions and definitions. The reader is often left guessing what the authors exactly meant or did or has to hunt through the manuscript to find the exact definition of a used term.

→ We thank Reviewer #1 for pointing this out. We have reformatted the ms. from that of Nature Methods (offering little space) to that of Nature Communications, which provides us with sufficient space to present terms, definitions and references in an adequate fashion.

2) PS-CFP2 is, compared to Eos or organic STORM dyes, very dim. This can make identifying a molecule in a given frame very unreliable since detection threshold or background levels have a major influence on the detection efficiency.

In the worst case, even small changes in background levels or threshold in the analysis algorithm can cause strong changes in the perceived blinking characteristics of PS-CFP2 (e.g. leading to completely different numbers of "gaps" or "detections"). This effect potentially renders the demonstrated simulation approach very unreliable. The authors should systematically investigate the robustness of their approach, for example how the data presented in Fig. 1E changes when data analysis parameters such as detection thresholds or experimental conditions (e.g. background or signal levels) change.

→ We respectfully disagree with Reviewer #1. Surely, single PS-CFP2 molecules require more excitation light to give rise to comparable signals than mEOS3.2 or red-shifted organic dyes. However, with the settings that we have chosen both for (i) establishing robust blinking statistics with the use of our platform and for (ii) imaging T-cells we arrive in all cases at clearly resolvable emission signals with a signal to noise ratio well above 30 – making it very unlikely that signals are missed by the detection algorithm. In

the current version we have dedicated a separate supplementary figure displaying the signal to noise statistics (**supplementary figure 2A**). As is evident and given that all data were acquired in TIRF configuration, signal to noise ratios remained comparable between experiments involving our customized platform. We have now investigated in separate experiments whether cells that do not express or are not decorated with PS-CFP2 / mEOS3.2 / Abberior CAGE 635 / PA Janelia Fluor 549 give rise to signals that are comparable to those recorded in fluorophore-positive cells. As shown in **supplementary figure 6C**, we have not found any, underscoring the overall integrity of our fluorophore detection (acquisition and image analysis).

The current statement "However, background fluorescence increased under these circumstances to levels which no longer allowed for a clear discrimination between localizations of single PS-CFP2 molecules and of background signals (Supplementary Fig. 2B)." (line 133) is in my view not sufficient.

→ In the current version we have rephrased and restructured the underlying message through quantitation of the increase in non-specific signals (**supplementary figure 5A-C**). We also point out an additional advantage that our platform offers, which is to assess the fidelity of fluorophore detection at the chosen settings.

3) Software code plays a major role in this manuscript. A major selling point of the manuscript is, that the method is readily applicable by the community. *The authors mention that their code is available for download, but do not provide any details where and how.* Similarly, no documentation of the code or a manual are provided. This should be added if the developed tool is supposed to have an impact.

→ We regret that Reviewer #1 did not get the opportunity to test our algorithms, which we had co-submitted as a ZIP file together with the text and figure files. To ensure that testing can be done at will, we have now provided a link (<https://owncloud.tuwien.ac.at/index.php/s/qnfepWzNLsZaf3H>) for downloading both the software and documentation on how to use it. After extensive discussions with potential software users we extended its functionality to support (i) the convenient determination of the blinking statistics as well as (ii) dual-channel registration.

4) Environmental conditions can influence the blinking behavior of fluorescent probes (as the authors state themselves for glutathione). Yet, for their T-cell experiments, they fix the cells, which was not done in the in vitro experiments, if I am not mistaken. It is unclear how this treatment compares to the experiments done with purified proteins which are used as the base of the simulations. *I suggest that the authors characterize PS-CFP2 under fixation conditions and use this data for the corresponding simulations.*

→ We thank Reviewer #1 for his/her thoughtful comment, which we fully agree with. In the current version we have determined blinking statistics for all four fluorophores investigated under conditions of formaldehyde-based cell fixation and applied those statistics for interpreting all localization maps involving cells fixed under the same conditions.

5) Page 10, Lines 206-209: The authors describe the observation of clusters when imaging antibody-labeled LFA-1 and explain this with clustering of LFA-1. *Another possibility would be that multiple biotins are bound to each antibody.* How did the authors rule out this possibility?

→ Again, we very much thank Reviewer #1 for his/her insightful and constructive comment. We have now determined the number of streptavidin-accessible biotin residues using a gel-shift assay (provided in **supplementary figure 6A and B**) and adjusted the biotinylation reaction to result in 1 biotin moiety per antibody (for more details on how we achieved this, please refer to Materials and Methods section).

6) A discussion of *possible artifacts arising from "undercounting"* (by not labeling every molecule or bleaching individual fluorescent proteins before they could be detected, for example) is completely missing. The authors should briefly explain why undercounting does not affect their analysis or how they addressed this issue.

→ We agree with Reviewer #1 that undercounting poses a potential problem to any cluster detection (especially clusters harboring a low number of molecules) which requires fluorophore labeling. While our method does not per se address fluorophore undercounting, it provides tools to assess the degree of undercounting. We would like to point out that our methodology is *per se* not affected by it. In the current version of the ms. we devote one paragraph to this issue in the Discussion section.

7) The discussion of the data summarized in Fig. 1E needs to include a statistical analysis to clarify if any of the observed changes are statistically significant.

→ We have now tested all data sets for statistical significance (supplementary table 1 and 2).

Additional minor comments:

- Line 103: "When applying an excitation power density of 3.0 kW/cm² for 2 ms, a single PS-CFP2 molecule was detected on average 3.47 times over the course of 10,000 frames" – what does the "2 ms" mean here? It should not be necessary to extract this kind of simple information from the methods section. Similarly, on page 6, line 104, it would be helpful to mention the wavelength and frame rate.

→ We have corrected this as was requested.

- Given the central importance of the term "detection" in this manuscript, it should be clearly and explicitly defined, which it currently is not. I interpret the different mentions (in particular, line 98 combined with Fig. 1D) to mean that a "detection" for the authors is the appearance of a molecule in one frame, and not that multiple occurrences of a molecule in one emission burst are merged into one "detection".

→ In the current version of the ms. we have more clearly defined the context in which we use the term "detection". Fluorophore detection refers to the appearance of a fluorophore in one frame. Observation of blinking of single molecules would hence involve the detection of a molecule in several frames. In contrast, cluster detection involves fluorophore detection and acquiring/employing fluorophore blinking statistics for evaluation.

- What is a "PS-CFP2 signal" (line 103)? It seems to be the trace of a single molecule since the number 1080 appears in the context of "signal" as well as the number of molecules in Fig. 1E. It would be helpful to not have the reader speculate about this by providing a clear definition and by using consistent terminology throughout the manuscript.

→ We have now rewritten this section with an emphasis on consistent term usage.

- On page 3, I suggest to include mMaple in the list of photoswitchable fluorescent proteins.

→ mMaple constitutes an interesting candidate to be tested for its blinking behavior with the use of our platform. We have now mentioned mMaple and cited its first publication in the current version of the ms..

- Page 5, lines 85-86: the authors call signals which only show green signal but no STAR635 signal "false positive". This terminology suggests that these signals represent "trace amounts of (partly) hydrophobic buffer- or lipid-derived dyes" (lines 74-75). It is, however, from this reviewer's perspective equally possible that STAR635 labeling efficiency of mSav* was far from 100% which would explain the large number of green-only signals as well. The authors should

discuss this possibility in their manuscript and either consider it in their later reasoning or clarify why they can rule out this possibility.

→ Given the methodology of protein labeling (maleimide chemistry under TCEP-mediated reducing conditions of monovalent streptavidin, a protein, which does not feature a native cysteine residue and which folds very efficiently in vitro under reducing, i.e. sulfhydryl-preserving conditions) and the protein:dye ratio, which we measured via photospectrometry, we consider it highly unlikely that protein labeling was not quantitative. We can however not fully exclude the possibility that a proportion of the purchased dye was non-functional and that the manufacturer's instructions/specifications were in part incorrect. In addition, heavy blinking of STAR635 precludes efficient detection. In any case, since this does not affect the integrity of our co-localization-based blinking analysis, we did not pay further attention to this possibility.

- The description of how Fig. 2A was generated (bottom of page 7) is very cryptic. I believe I have figured it out after quite some thought, but I am still not 100% sure that I got it. This can probably be fixed by more careful phrasing and consistent use of definitions. For example, what does "sorting the detections per molecule in ascending order" (line 145) mean? Do the authors mean "sorting the molecules by their number of detections"? Are "fluorophore localizations" (line 146) the same as "detections"? Overall, it is not clear what information Fig. 2A provides that is not already provided in Fig. 1E.

→ We thank Reviewer #1 for pointing this out. We have now added a description to the Materials and Methods section to be found more intelligible. Figure 2A (now **figure 2E-H**) allows for a direct comparison of statistics for different settings applied for one fluorophore.

- First paragraph on page 8: Why are the maxima of the plots in Fig. 2A not exactly equal to the mean number of detections per molecule?

→ We thank Reviewer #1 for this comment. We have now entirely revised figure 2A (now **figure 2E-H**) and arrive at maxima which are equal to the mean number of detections per molecule.

- What are the "blinks" mentioned in Fig. 2B? Are "detections" in subsequent frames combined into one emission event (as it is usually done), or not? This needs to be declared at least, or better, both images (with and without the combination of subsequent detections into one event) should be displayed.

→ We have now exchanged "blinks" with "ground truth including blinking".

- Given the importance of the confidence intervals for Ripley's K function in the developed method, it would be helpful to explicitly describe how they are obtained. It seems that it is the S.D. of 15 simulations, but for what field of view and what number of molecules? One can collect it from little pieces of information throughout the manuscript, but a clear statement where the confidence intervals are first introduced would lower the frustration of readers and avoid misunderstandings.

→ We thank Reviewer #1 for pointing this out. We have now clearly indicated how we obtained the reported confidence intervals (indeed standard deviation derived from 15 simulations involving 70 molecules per square microns in an area of 15 x 15 μm^2).

- Fig. 2D: why is the number of clusters per μm^2 used a relevant parameter? It seems to me that the total number of molecules would be a more relevant parameter, since the number of clusters per μm^2 makes only sense for a particular field of view and therefore makes it difficult to compare between microscopes.

→ Here, we respectfully disagree with Reviewer #1. For simulations we used an area-independent density of 70 molecules per μm^2 . A certain fraction of these molecules was assigned to reside in clusters ("%" of

molecules in clusters”) with a certain radius (“cluster radius”). By defining a cluster density, the absolute number of molecules inside or outside can be directly calculated without any further knowledge of the overall area. The only reason for using larger areas than $1\mu\text{m}^2$ in simulations is the gain in statistics for the determination of Ripley’s K functions. Values for the number of clusters were chosen based on our previous studies about TCR clustering on the nanoscale (Rossboth et al., *Nature Immunology* 2018).

Reviewer #2 (Remarks to the Author):

The manuscript from Rossoboth et al. addresses the issue of molecular blinking of PSCFP2 and its influence in proper molecular counting and cluster detection in the context of PhotoActivated Localization Microscopy. The authors perform most of their experiments in vitro, and provide in the end an application in fixed T cells. The findings are then generalized to propose a pipeline to use blinking information to sort in a rigorous way clustering estimator (such as Ripley's K/L functions) to determine to what extent a sample is displaying true oligomerization or artifactual clustering due to repeated counting of the same molecule.

→ We thank Reviewer #2 for his/her valuable time and efforts needed to evaluate our ms. and for his/her constructive criticism.

I have several major concerns with this manuscript.

The first and foremost, is that the authors address a question that has been around since about a decade now (see Lee et al. PNAS 2012, Annibale et al JPCL 2010), focusing on a first-generation fluorescent protein, PS-CFP2, which has also been used since 2007. Since then, a large number of reports (most recently, and to cite only a few, Fricke et al. Sci Rep 2017, Rollins et al. PNAS 2014, Dursic et al. Nat. Methods 2014) have addressed this issue proposing a wide array of methods to characterize photoblinking/single molecule photophysics and include it in the resulting image analysis process in order to avoid counting/clustering artifacts. Given the crowded field, it is not clear what are (i) the novelty and (ii) the generality of the findings and of the approach proposed by the authors.

→ We fully agree with Reviewer #2 that the central question addressed in our ms. has been dealt with by others for as long as PALM and STORM are in use. We would even like to go a step further and state that despite previous reports, blinking has remained to this day a lingering issue in superresolution microscopy which significantly limits the impact of its use. This emphasizes rather than diminishes the importance of what we report in our ms..

We speak from experience: as T-cell biologists focusing on a deeper mechanistic understanding of how T-cells manage to detect with their low (micromolar) affinity T-cell antigen receptors the presence of even a single (!) antigen on the surface of antigen presenting cells, we surely perceive the importance and paucity of reliable strategies to detect real molecular clusters with high sensitivity. Truth is, they do not exist, and their absence has caused much confusion and wasted efforts in the field with profound implications affecting studies on the etiology of autoimmune diseases, lack of immune surveillance in cancer and for designing more effective T-cell based immunotherapies.

To our knowledge, we are the first to provide a sophisticated solution to address blinking-associated parameters such as on- and off-times, which, as we have shown, are critical for cluster assessment. The reason for this is that our imaging platform affords at current the only available approach which allows for reliable extraction of such parameters due to the nature of single molecule analysis and the co-localization-based strategy to identify single photo-switchable and photo-activatable fluorophores. When combined, both aspects of our methodology ensure detailed characterization of blinking behavior that is relevant for cluster evaluation. This has enabled us to identify outliers in on- and off-times distributions, which cannot be accounted for by event merging or fitting of respective distributions, which is suggested as an effective measure in previous studies (see e.g. Annibale et al., PLoS One 2011, Lee et al., Proceedings of the National Academy of Sciences 2012, Fricke et al., Scientific Reports 2015, Hummer et al., Mol Biol Cell 2016, Puchner et al., Proceedings of the National Academy of Sciences 2013 or the theoretical work by Rollins et al., Proceedings of the National Academy of Sciences 2015).

Furthermore and equally important, our platform is accessible to many life scientists from different fields using PALM, who are less firm in the field of photochemistry (e.g. biochemists, biologists, immunologists, neurobiologists etc..) as they have now a tool to identify fluorophores and to determine imaging conditions that are best suited for their biology at hand.

While one may argue that PS-CFP2 has so far been considered a largely non-blinking fluorophore and any information on its true blinking behavior should be disseminated, nevertheless, the current manuscript unfortunately does not focus on providing an exhaustive spectroscopic characterization of the behavior of PS-CFP2.

→ While such undertaking may be worthwhile in its own right, we do not understand the merit of it in the context of our ms.. A full spectroscopic characterization as suggested by Reviewer #2 would require substantially more resources and time yet without added value for the task at hand, which concerns the interpretation of localization maps derived from PALM imaging. After all, we do not pretend to offer a comprehensive photophysical characterization of any photo-switchable fluorophore but instead showcase a platform which allows for optimization of imaging conditions and robust evaluation of localization maps.

The photophysical characterization of the fluorophore behavior in the current manuscript falls short of the state of the art. When the reversible photo switching behavior of Dronpa, a reversible photoswitching fluorescent protein, was first investigated (Habuchi et al. PNAS 2005), the authors measured Dronpa on a confocal setup, using Avalanche Photodiodes: this yields the sub ms-temporal resolution necessary to obtain a credible photophysical model for the behavior of the fluorophore.:

→ Again, and in line with what we have stated above, it is not clear to us why such time-resolved characterization would provide any added value to our platform. The rationale that we follow is to adapt the characterization of blinking to the needs defined by our imaging methodology. We would like to emphasize that the sole purpose of acquiring the blinking statistics for any given fluorophore at any given imaging conditions is to be able to interpret localization maps generated for cells imaged under the very same conditions. Understanding in detail the photophysical basis underlying blinking is without doubt a valuable endeavor for e.g. choosing experimental parameters optimized for super resolution experiments, (see e.g. the detailed characterization of Dendra2 by the Hess lab, Pennacchietti et al., Biophysical Journal 2017), but clearly outside the scope of our ms.. Our argument is also supported by the reviewers' comment about the characterization of Dronpa: while single molecule localization microscopy requires the illumination of a large sample area and the use of sensitive sCMOS or (EM)CCD detectors operating on a milliseconds timescale, fast avalanche photodiodes and a confocal microscopy system are needed for a detailed photophysical characterization of fluorophores. Our methodology allows the use of the same microscopy system for the characterization of fluorophores and their subsequent use in SMLM experiments.

A number of questions about CFP2 blinking thus remain unanswered: What is the role of pH in determining the blinking behavior of PS-CFP2?

→ We thank Reviewer #2 for this comment as pH affects protein conformation and consequently its photophysics and blinking behavior. We have therefore focused our analysis on fixed cells which rapidly equilibrate after fixation to pH values found within the imaging buffer. Since we employ the same imaging buffer both in our platform and our cellular imaging studies, pH values are identical and hence represent no longer a concern. However, the reviewer addresses an interesting point which can be more generalized: how does the local environment or experimental parameters change the fluorophore's blinking behavior. We now provide blinking characterization of fluorophores under various conditions, which show a minor dependence of blinking on the presence of PFA, slightly reducing conditions and changes in excitation power.

Furthermore, only two power levels (a third is deemed to generate too high background) are used to characterize the change of the blinking parameters. Is this sufficient to extract a meaningful photophysical model for PSCFP2, as done in Fig. 5 of Habuchi et al? Can different combinations of photoactivation (405 nm) and excitation (488 nm) light

contribute to modulate the blinking pattern? What about the behavior of the fluorophore in the Cyan, non-photoconverted form?

→ Again, these are all interesting questions. However, we do not see how answers to these questions will affect the outcome and integrity of our overall approach. Obtaining them is hence outside the scope of our study.

Finally, it is not clear how the pipeline which the authors provide, namely to categorize Ripley's function by taking into account the 'blinking fingerprint' of the fluorophores, can be generalized to other fluorescent proteins, and possibly, to other dyes, such as those used in STORM. Since 2006, and the development of PALM and STORM, the two techniques have become somehow interchangeable: STORM being favored for the higher brightness of the organic dyes used and the typically higher resolution of the reconstructed sub-diffraction limited structures. However, given the stochastic, rapid blinking of STORM dyes, the possibility to count molecules in STORM (see Finan et al. Angewandte 2015) is much more challenging than with fluorescent proteins. For any claim of generality, my recommendation to the authors would be to definitely expand their investigation to other FPs and also organic dyes.

→ We fully agree with Reviewer #2 that showcasing more than one fluorophore helps to support our claim that our platform is universally applicable. In the revised version of our ms. we have therefore expanded the analysis to include in total four photo-switchable/photo-activatable fluorophores: the two fluorescent proteins PS-CFP2 and mEOS3.2 and the two organic dyes Abberior CAGE 635 and PA Janelia Fluor 549. As becomes evident from our analysis, PS-CFP2 featured the least blinking and gave in turn rise to the highest sensitivity in cluster detection, while Abberior CAGE 635 showed the most pronounced blinking. Interestingly, PA Janelia Fluor 549, which has been reported to feature much reduced blinking, shows a behavior similar to PS-CFP2.

Of note, we refrain from making any recommendations for STORM, as the underlying photophysical and photochemical processes are highly heterogeneous and render data interpretation too complex to be adequately accounted for by any methodology based on blinking fingerprints.

Other Major comments:

The authors in Fig. 1 provide two representative traces, as well as statistics based on approximately 1000 molecules. Given the sophisticated single molecule imaging platform which they use, one would expect it would be possible to harvest a significant larger amount of data. Furthermore, if the point is that outlier's behavior does matter, then a 1% outlier population is represented here by only about 10 molecules, which is a rather limited statistical sample.

→ Contrary to Reviewer #2' guess, there is considerable work involved in acquiring enough data to support robust statistics: To ensure single molecule resolution we employed fluorophores at densities amounting to about **5 to 10 molecules per field of view** (about $12 \times 24 \mu\text{m}^2$), which were then imaged for 10,000 frames at a frame rate of **167 (53) Hz** (at 4 (0.4) kW cm^{-2}) (**total duration ~ 2 (4) minutes per run**). To arrive at a sufficient number of events we repeated such runs up to **150** times for each fluorophore and imaging condition.

We agree with the Reviewer, that 1% of outliers would not involve a lot of molecules, however, it is not only individual outliers which matter but the extremely long tail observed e.g. for the off-time distribution of PS-CFP2. If the distribution is described by a simple exponential function, as was used for several counting approaches based on blinking, about ~18% of off-time "outliers" would be missed and ~7% of multiple detection events.

The authors propose a 'comprehensive methodology' to discriminate datasets based on comparing Ripley's functions, as done in Fig. 2c-d. I.e., generating Ripley's functions of spatially random datasets, affected by the blinking behavior. Previous work has however been done along these lines (Shivanandan et al, PlosOne 2015). In addition, one would expect that Ripley's functions' behavior to be affected also by factors like the localization error of

the fluorophore, as well as the total number of fluorophores sampled, i.e. the local sample density, therefore unless these cases are explicitly discussed, it becomes unclear to determine how quantitatively reliable the comparison of Ripley's functions of heterogeneous samples to simulated random controls can be in assessing their effective degree of clustering.

→ We fully agree that the shape of Ripley's K function depends on the local density. For Monte Carlo simulations we first determine the local density based on the provided local localization map and the average number of blinks per molecule for a given fluorophore. We included details concerning our methodology in the software documentation, which was initially submitted to the journal but unfortunately not sent to the reviewers. We now also mention the density determination explicitly in the main ms. article file. This holds also true for the localization error, a parameter which can be easily adjusted to match experimental results.

The paper by Shivanandan et al., PLoS One 2015 describes the influence of detection efficiency and localization error on the cluster size determined via Ripley's K analysis. The method – as many others – uses Monte Carlo simulated data to finally determine the corrected size of clusters. While this method may be adequate when dealing with well defined and uniform cluster sizes, our 'comprehensive methodology' reports instead on the mere presence or absence of clustering. To render the 'comprehensive' context of our method more apparent, we now provide an extended software package which allows to (i) register color channels, (ii) determine blinking statistics and (iii) classify Ripley's K function to distinguish molecular clustering from random distributions.

REFERENCES

1. P. Annibale, S. Vanni, M. Scarselli, U. Rothlisberger, A. Radenovic, Quantitative photo activated localization microscopy: unraveling the effects of photoblinking. *PLoS One* **6**, e22678 (2011).
2. S.-H. Lee, J. Y. Shin, A. Lee, C. Bustamante, Counting single photoactivatable fluorescent molecules by photoactivated localization microscopy (PALM). *Proceedings of the National Academy of Sciences* **109**, 17436-17441 (2012).
3. F. Fricke, J. Beaudouin, R. Eils, M. Heilemann, One, two or three? Probing the stoichiometry of membrane proteins by single-molecule localization microscopy. *Scientific Reports* **5**, 14072 (2015).
4. G. Hummer, F. Fricke, M. Heilemann, Model-independent counting of molecules in single-molecule localization microscopy. *Mol Biol Cell* **27**, 3637-3644 (2016).
5. E. M. Puchner, J. M. Walter, R. Kasper, B. Huang, W. A. Lim, Counting molecules in single organelles with superresolution microscopy allows tracking of the endosome maturation trajectory. *Proceedings of the National Academy of Sciences* **110**, 16015-16020 (2013).
6. G. C. Rollins, J. Y. Shin, C. Bustamante, S. Pressé, Stochastic approach to the molecular counting problem in superresolution microscopy. *Proceedings of the National Academy of Sciences* **112**, E110-E118 (2015).
7. F. Pennacchietti, T. J. Gould, S. T. Hess, The Role of Probe Photophysics in Localization-Based Superresolution Microscopy. *Biophysical Journal* **113**, 2037-2054 (2017).
8. A. Shivanandan, J. Unnikrishnan, A. Radenovic, Accounting for limited detection efficiency and localization precision in cluster analysis in single molecule localization microscopy. *PLoS One* **10**, e0118767 (2015).

REVIEWERS' COMMENTS:

Reviewer #1 (Remarks to the Author):

The authors have addressed all of my concerns raised in the previous review. Except for a few minor concerns which should be easily and quickly addressable, I consider the manuscript acceptable for publication in Nature Communications.

Minor concerns:

- Line 24-26: This sentence is confusing. Do the authors mean "photobleaching" instead of "photoswitching"? Consider revising.
- Line 138-140: If I interpret this sentence correctly, only the average number of detections of all the measured properties changed in a statistically significant manner, and did so only for the two organic dyes but not the two fluorescent proteins. I not see how Suppl. Table 1 supports this claim. Do I misinterpret the sentence? Is there a typo in the table? Please clarify.
- Page 8: related to the previous comment, I am slightly concerned to interpret the absence of a significant difference ($p > 0.05$) as evidence that two parameters are "largely unchanged" (e.g. line 149). A p-value of larger than 0.05 can also be the result of insufficient precision of an experiment or an insufficient sample size. I recommend that the authors consider revising the text slightly throughout primarily this section of the manuscript keeping this concern in mind.
- Line 161-162: This sentence is ambiguous. What do the authors mean by "continuous decrease"? "monotonic decrease"? And why is this a confirmation? If it cannot easily be clarified, it might be better to delete this sentence since it does not add much additional information, in my opinion.

Reviewer #2 (Remarks to the Author):

In the revised version of their manuscript Platzter et al. address only partially the concerns initially raised by the reviewers. In particular, three other fluorophores have been added to the analysis (PS-CFP2 still remains the 'best' candidate for blinking-free cluster detection). However, key concerns about a more thorough spectroscopic characterization of the fluorophore initially chosen for the study, PS-CFP2, have been largely dismissed. Furthermore, the issues initially raised with respect to the limited spectroscopic characterization of PS-CFP2, now also apply to the three new fluorophores.

As also pointed out by Reviewer 1, there are a number of instances where the environmental conditions can affect the behavior of these fluorophores. Without a more quantitative understanding of their photo-physics the study remains largely observational.

There is an intimate relationship between the very complex photophysical behavior of fluorophores and their performance in single molecule localization microscopy. My encouragement as a reviewer in the first round, was to generate as much data as possible on this aspect, in order to provide not just a black-box method that could be applied to specific fluorophores in given experimental conditions, but rather a map that could help guiding others to navigate the general problem.

If the suggestion of the authors is that one should use Ripley's functions and PSCFP2 to study clustering, then they should also show how this analysis may change when facing the broad spectrum of possible experimental conditions faced when setting up such an experiment.

Two egregious examples:

1. upon the suggestion to look at the pH dependence of the probes, the authors reply "We have therefore focused our analysis on fixed cells which rapidly equilibrate after fixation to pH values found within the imaging buffer." But this is clearly not the point. How would this help a scientist who is interested in working in living cells, perhaps with specific media and buffer? There is no

mechanistic information in this study that could help him/her in determining what properties of his buffer of choice could affect the blinking behavior of the probe.

The limited additional information provided in Figure 2 E-H would not help either, as it does not contribute to provide a photophysical model for the fluorophore. I would point the authors to the excellent work by Bizzarri et al (doi/10.1021/ja9014953), which actually allows some predictions to be made about how environmental or spectroscopic changes may affect the switching behavior of the fluorophore.

2. Suggestion to look at the effect of combining different excitation and photoactivation power levels are deemed "outside the scope of the study". How can this be out of the scope of the study? If the authors suggest that they are only providing an analytical tool (based on long-established Ripley's K functions) to characterize clustering, and that the determination and knowledge of a photophysical model of the fluorophore used has ultimately no bearing on the outcome of such cluster analysis, perhaps they should consider a more specialized publication.

Finally, the authors explicitly exclude STORM (accounting for a significant fraction of the localization microscopy studies up to date, included those on clustering) from the techniques which could benefit from their method. This time, the reason is that the "photophysical and photochemical processes are highly heterogeneous". This statement appears at odds with the claim that generating a photophysical model for the fluorophore used is "outside the scope of our study".

AUTHORS' POINT-BY-POINT RESPONSE TO THE REVIEWERS' COMMENTS

The reviewers' comments are written in Arial Narrow font type. Our responses are introduced with an arrow (→) and set in brick red Times New Roman.

Reviewer #1:

The authors have addressed all of my concerns raised in the previous review. Except for a few minor concerns which should be easily and quickly addressable, I consider the manuscript acceptable for publication in Nature Communications.

→ We thank Reviewer #1 for his/her time to study also our revised work in much detail and to point out unclear sentences which we have now amended.

Minor concerns:

- Line 24-26: This sentence is confusing. Do the authors mean "photobleaching" instead of "photoswitching"? Consider revising.

→ We agree with the reviewer, that "Photoswitching reduces (...)" might be misleading for some readers and have changed the corresponding sentence to "Photoswitching only a minor fraction of fluorophores into the active state gives rise to well-separated single molecule signals, which are localizable with a precision primarily determined by the signal to noise ratio".

- Line 138-140: If I interpret this sentence correctly, only the average number of detections of all the measured properties changed in a statistically significant manner, and did so only for the two organic dyes but not the two fluorescent proteins. I [do] not see how Suppl. Table 1 supports this claim. Do I misinterpret the sentence? Is there a typo in the table? Please clarify.

→ We agree with the reviewer – our determined significance values in Suppl. Table 1 do not support the statement provided. In the revised version we have removed the unsupported claim.

- Page 8: related to the previous comment, I am slightly concerned to interpret the absence of a significant difference ($p > 0.05$) as evidence that two parameters are "largely unchanged" (e.g. line 149). A p-value of larger than 0.05 can also be the result of insufficient precision of an experiment or an insufficient sample size. I recommend that the authors consider revising the text slightly throughout primarily this section of the manuscript keeping this concern in mind.

→ We have changed lines 149 and 273 according to the reviewer's comment.

- Line 161-162: This sentence is ambiguous. What do the authors mean by "continuous decrease"? "monotonic decrease"? And why is this a confirmation? If it cannot easily be clarified, it might be better to delete this sentence since it does not add much additional information, in my opinion.

→ We agree with the reviewer that the "monotonic decrease" is less important compared to observation of the very broad t_{off} distribution due to appearance of background signals at random time points. We changed the sentence accordingly.